# Free-standing two-dimensional ferro-ionic memristor

Jinhyoung Lee[1,2,9], Gunhoo Woo[3,4,9], Jinill Cho[1], Sihoon Son[3,4], Hyelim Shin[5], Hyunho Seok[3,4], Min-Jae Kim[3,4], Eungchul Kim[6], Ziyang Wang[1], Boseok Kang ⓘ [3,4,7], Won-Jun Jang[2,8] & Taesung Kim ⓘ [1,3,4,5,7] ✉

Two-dimensional (2D) ferroelectric materials have emerged as significant platforms for multi-functional three-dimensional (3D) integrated electronic devices. Among 2D ferroelectric materials, ferro-ionic $CuInP_2S_6$ has the potential to achieve the versatile advances in neuromorphic computing systems due to its phase tunability and ferro-ionic characteristics. As $CuInP_2S_6$ exhibits a ferroelectric phase with insulating properties at room temperature, the external temperature and electrical field should be required to activate the ferro-ionic conduction. Nevertheless, such external conditions inevitably facilitate stochastic ionic conduction, which completely limits the practical applications of 2D ferro-ionic materials. Herein, free-standing 2D ferroelectric heterostructure is mechanically manipulated for nano-confined conductive filaments growth in free-standing 2D ferro-ionic memristor. The ultra-high mechanical bending is selectively facilitated at the free-standing area to spatially activate the ferro-ionic conduction, which allows the deterministic local positioning of $Cu^+$ ion transport. According to the local flexoelectric engineering, $5.76 \times 10^2$-fold increased maximum current is observed within vertical shear strain 720 nN, which is theoretically supported by the 3D flexoelectric simulation. In conclusion, we envision that our universal free-standing platform can provide the extendable geometric solution for ultra-efficient self-powered system and reliable neuromorphic device.

Two-dimensional (2D) van der Waals ferroelectric materials, which exhibit non-volatile switching with switchable polarization, have emerged as promising platforms for next-generation functional electronics, including the energy storage[1–4], field-effect transistor[5–7], ferroelectric tunneling junction[8,9], bulk photovoltaic effects[10–12], and neuromorphic computing[13–15]. Among such 2D ferroelectric materials, ferro-ionic $CuInP_2S_6$ has received considerable attention due to its

phase-tunable characteristics[16–18]. The phases can be classified as (i) ferroelectric phase, (ii) ferro-ionic phase, and (iii) paraelectric phase, and correlated with interlayer $Cu^+$ ion dynamics[19]. In addition, $CuInP_2S_6$ has intrinsic mechanical flexibility with a scalable 2D van der Waals structure[20], which enables sensitive detection of external mechanical strain[21,22]. Depending on the tunable $Cu^+$ ion dynamics and mechanical flexibility, $CuInP_2S_6$ has the potential to enable versatile

[1]School of Mechanical Engineering, Sungkyunkwan University (SKKU), Suwon-si, Gyeonggi-do 16419, Republic of Korea. [2]Center for Quantum Nanoscience, Institute for Basic Science (IBS), Seoul 03760, Republic of Korea. [3]SKKU Advanced Institute of Nanotechnology (SAINT), Sungkyunkwan University, Suwon-si, Gyeonggi-do 16419, Republic of Korea. [4]Department of Nano Science and Technology, Sungkyunkwan University, Suwon-si, Gyeonggi-do 16419, Republic of Korea. [5]Department of Semiconductor Convergence Engineering, Sungkyunkwan University, Suwon-si, Gyeonggi-do 16419, Republic of Korea. [6]AVP Process Development Team, Samsung Electronics, Chungcheongnam-do Cheonan-si 31086, Republic of Korea. [7]Department of Nano Engineering, Sungkyunkwan University, Suwon-si, Gyeonggi-do 16419, Republic of Korea. [8]Department of Physics, Ewha Womans University, Seoul 03760, Republic of Korea. [9]These authors contributed equally: Jinhyoung Lee, Gunhoo Woo. ✉e-mail: tkim@skku.edu

advances in neuromorphic computing systems through switchable ferro-ionic conduction[18,23–26].

However, 2D ferro-ionic $CuInP_2S_6$ has major challenges in terms of commercialization and practical applications. As $CuInP_2S_6$ exhibits a ferroelectric phase with insulating characteristics at room temperature[16], external conditions such as temperature[27,28] and electrical field[29,30] are required to facilitate the $CuInP_2S_6$ phase transition. However, the external electric field and temperature inevitably induce material degradation and electrical instability of $CuInP_2S_6$[31]. Moreover, such external conditions simultaneously facilitate the stochastic conductive filament formation with randomized $Cu^+$ ion transport, which completely limits the device-to-device reliability and reproducibility of 2D ferro-ionic memristors[32].

Recently, mechanical strain has been suggested as a reliable ferroelectric domain switching method for various ferroelectric materials[33,34], which effectively hinders dielectric breakdown, leakage current, and stochastic ion migration. When a local inhomogeneous strain is applied to ferroelectric materials through the tip of an atomic force microscope (AFM), the mechanical bending induces a flexoelectric field[35,36], which is derived from the net charge shift in the lattice unit cell. As the mechanical bending is inversely proportional to the length scale, the flexoelectric field exhibits dominant effects in nanoscale materials[37]. Nevertheless, the mechanical switching of $CuInP_2S_6$ has proved challenging owing to dimensional limitations[38,39], due to the incompatibility of film thickness variation and insufficient local flexoelectric field for the $CuInP_2S_6$ phase transition[38]. These intrinsic physical limitations of $CuInP_2S_6$ have motivated us to develop an on-demand free-standing axial nanogap platform.

In this study, we present a programmable flexoelectric engineering platform for nanoconfined conductive filaments in free-standing 2D ferro-ionic memristor, which provides a geometrical solution for conventional stochastic limitations[40,41] and structural limitations[38,39]. To mechanically achieve the $CuInP_2S_6$ phase transition and ultra-efficient flexoelectricity, an axial nanogap structure was fabricated to locally induce free-standing $CuInP_2S_6$ at the nanogap. Owing to the flexibility of 2D van der Waals materials, the local inhomogeneous shear strain in the free-standing area allowed for an ultra-high strain gradient, unlike conventional ionic memristors with inflexible oxide materials. As a result, $5.76 \times 10^2$-fold maximum current ($I_{max}$) enhancement and $6.25 \times 10^2$-fold increased $I_{max, with strain} / I_{max, without strain}$ ratio was experimentally observed with vertical shear strain 720 nN with 25 nm tip radius. Since the upward polarized bottom $In_2Se_3$ suppresses the undesirable ionic conduction in the suspended junction area, a topographical $Cu^+$ ion extraction was selectively derived within 232.42 nm width and 58.98 nm height. In conclusion, this study proposes a previously unreported flexoelectric approach for nanoscale $Cu^+$ ion localization in free-standing 2D ferro-ionic memristor, which effectively permits reliable neuromorphic computing and ultra-low energy consumption systems.

## Results and discussion

### Free-standing axial nanogap platform for spatial ferro-ionic nano-manipulation

To achieve the confined growth of conductive filaments and spatial controllability of $Cu^+$ ion transport, we adopted a free-standing 2D ferro-ionic memristor with an axial nanogap platform, as shown in Fig. 1a. As shown in Fig. 1b, the free-standing 2D ferro-ionic memristor consisted of an $Au/\alpha$-$In_2Se_3/CuInP_2S_6/Pt$ structure, which was

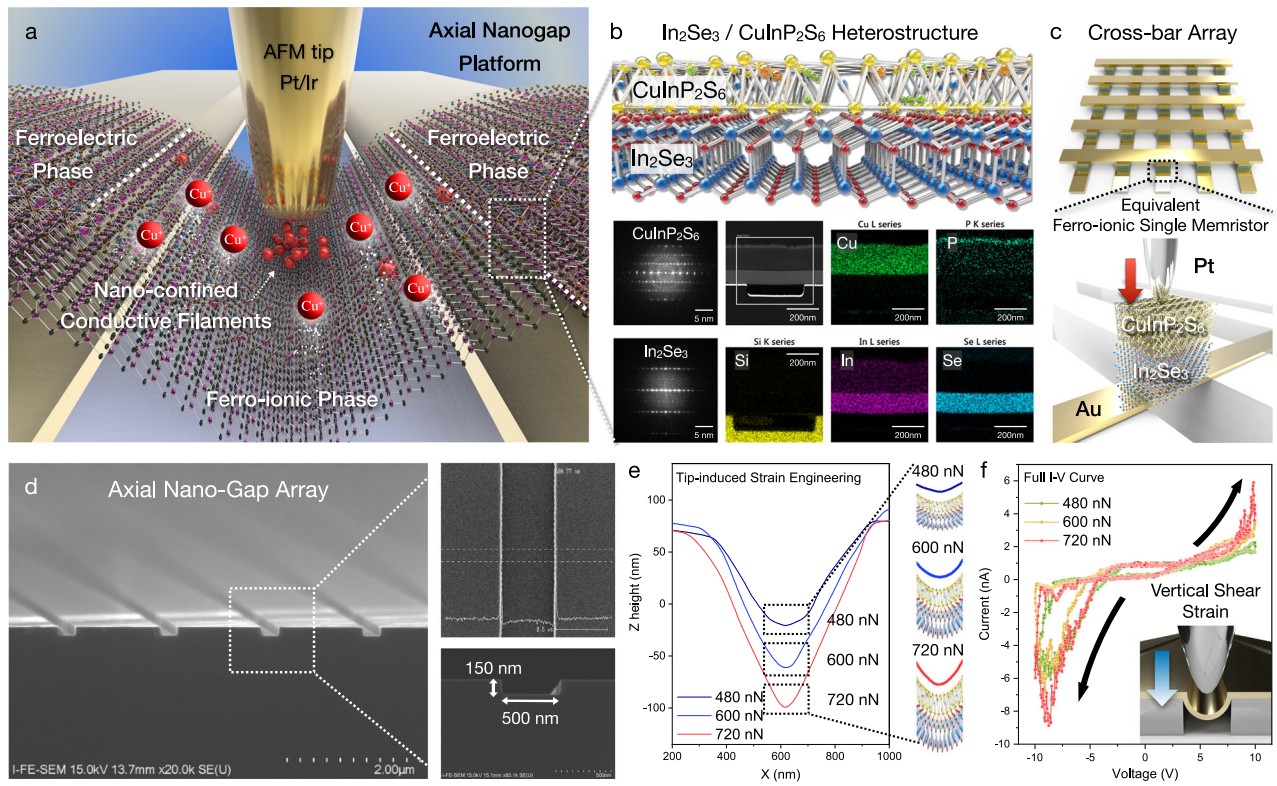

**Fig. 1 | Universal axial nanogap platform for programmable flexoelectric engineering. a** Schematic of a free-standing single 2D ferro-ionic memristor for spatial confinement of conductive filaments. **b** Free-standing 2D $\alpha$-$In_2Se_3/CuInP_2S_6$ heterostructure and its corresponding TEM cross-sectional image and SAED mapping. **c** Schematic illustration of the 2D ferro-ionic memristor structure, constructed with an Au bottom electrode/$\alpha$-$In_2Se_3/CuInP_2S_6/Pt$ top electrode. **d** SEM image of the axial nanogap array with a 500 nm width and 150 nm depth. **e** Cross-sectional 3D line profile with tip-induced strain engineering. Flexoelectric current amplification effects in the **f** full $I–V$ curve through shear strain.

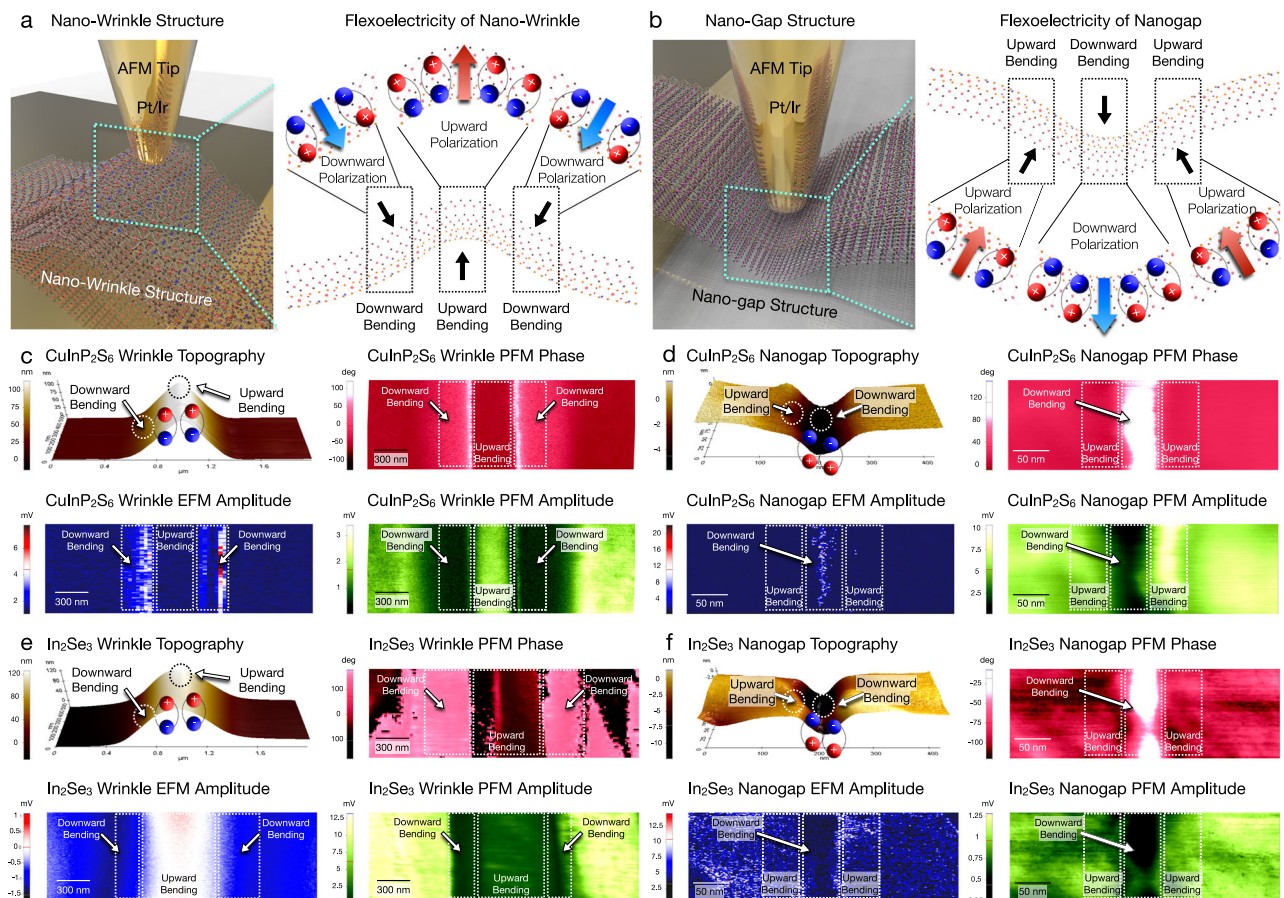

**Fig. 2 | Local flexoelectric domain mapping with local strain distribution.** Schematic of the local strain distribution and its corresponding flexoelectric field in the **a** nanowrinkle and **b** nanogap. Spatial domain imaging of the **c** nanowrinkle $CuInP_2S_6$, **d** nanogap $CuInP_2S_6$, **e** nanowrinkle $In_2Se_3$, and **f** nanogap $In_2Se_3$. Owing to the local flexoelectric polarization, PFM measurements indicate the downward polarization, as observed in the PFM amplitude and PFM phase. The electron distribution was correlatively obtained for the ferroelectric insulator $CuInP_2S_6$ and ferroelectric semiconductor $In_2Se_3$, which originated from the heterogeneous carrier density of the semiconductor and insulator.

fabricated with the dry transfer method at the axial nanogap structure (Supplementary Figs. 1 and 2). Cross-sectional transmission electron microscopy (TEM) and corresponding energy dispersive spectroscopy (EDS) element mapping (Fig. 1b) were conducted to visualize the device structure of the 2D ferro-ionic memristor, which constructed with an Au bottom electrode (20 nm), $\alpha$-$In_2Se_3$ (125.37 nm), $CuInP_2S_6$ (197.91 nm), and Pt tip as the top electrode. Additionally, device construction of 2D ferro-ionic memristor was exhibited with 3D schematic illustration for structural clarification and 3D stacking order (Fig. 1c). As illustrated in Fig. 1c (top), the memristor has been organized at the crossbar arrays to enable the mechanical bending within conductive AFM tip (Pt), which can act as a top electrode and strain source, was vertically manipulated as illustrated in Fig. 1c (bottom). To achieve the spatial modulation of ferro-ionic conduction, mechanical switching has recently emerged as a versatile ferroelectric domain switching method. Nonetheless, the previous research of flexoelectric engineering was completely limited due to the geometrical substrate suspension[23] and anisotropic mechanical bending[38], while the axial nanogap platform (Fig. 1d) derives the local free-standing state and an efficient mechanical bending, unlike the conventional structural platform. As shown in Fig. 1e, the nanoscale mechanical bending in the free-standing state was precisely controlled with vertical shear strain. The nanoscale bending curvature was independently manipulated at the heterogeneous free-standing gap region through the tip-induced vertical shear strain. The nanoscale bending curvature radius was experimentally obtained as 97.39 nm (480 nN), 38.72 nm (600 nN), and 25.14 nm (720 nN). Following the tip-induced strain engineering,

3D topography was measured with conventional AFM imaging. Thus, the separation of tip-induced strain engineering and topography scanning enables the accurate topographical imaging in Fig. 1e. While the thickness of free-standing 2D ferroelectric heterostructure has been measured as 214.04 nm, the naturally-induced mechanical bending at the nanogap was hindered unlike monolayer and bilayer scale, which requires the external force (478.32 nN) to achieve nanoscale bending curvature at the nanogap region. The obtained mechanical bending induced downward (upward) polarization and net polarization shift in the lattice unit cell, and hence, a gradual current amplification was clearly observed under the nanoscale flexoelectric engineering, as shown in Fig. 1f.

## Flexoelectric domain switching behavior via nanoscale axial mechanical bending

To specifically realize flexoelectric engineering with the local strain distribution, the heterogeneous flexoelectric behavior of the downward bending and upward bending was spatially mapped within the nanogap and nanowrinkle structures through correlative electrostatic force microscopy (EFM) and piezoelectric force microscopy (PFM) imaging after the initial +7.5 V sample bias poling. Before the flexoelectric domain mapping, pre-characterization of single $CuInP_2S_6/\alpha$-$In_2Se_3$ flake was performed with PFM imaging and XPS measurements (Supplementary Fig. 3). Figure 2a, b indicates the downward bending locally generated downward polarization in the nanogap center area and the nanowrinkle slope area. Conversely, upward polarization was localized by the upward bending occurring in the nanogap slope area

and nanowrinkle center area. When the inhomogeneous strain induced the nanoscale lattice bending, which locally activates the internal flexoelectric field in the material, we observed local polarization switching and carrier redistribution. The nanoscale heterogeneity of the ferroelectric insulator $CuInP_2S_6$ and ferroelectric semiconductor $\alpha$-$In_2Se_3$ were probed with their structural variations, as shown in Fig. 2c–f. As the $CuInP_2S_6$ mechanical bending caused upward and downward polarization, respectively, the ferroelectric domain was partially engineered owing to the flexoelectric polarization direction. Heterogeneous piezoresponse force microscopy (PFM) amplitude and phase at the edge and center areas were observed to be 11.37 mV / 159.89° ($CuInP_2S_6$ nanogap), 1.04 mV / 111.84° ($\alpha$-$In_2Se_3$ nanogap), 55.03 mV / 103.90° ($CuInP_2S_6$ nanowrinkle), and 11.37 mV / 356.30° ($\alpha$-$In_2Se_3$ nanowrinkle). To further clarify the heterogeneous electron distribution, the EFM amplitude at the edge and center area were spatially mapped as −21.70 mV ($CuInP_2S_6$ nanogap), 13.76 mV ($\alpha$-$In_2Se_3$ nanogap), 6.76 mV ($CuInP_2S_6$ nanowrinkle), and −2.75 mV ($\alpha$-$In_2Se_3$ nanowrinkle), which indicates the nanoconfined electron distribution within inhomogeneous strain, attributed to the down polarization. In contrast, the $\alpha$-$In_2Se_3$ EFM amplitude exhibits confined hole distribution in the downward bending area, corresponding to the semiconductive properties of $\alpha$-$In_2Se_3$, which can be further correlated with heterogeneous EFM phase shift (Supplementary Fig. 4). During the tip-induced strain engineering, the homogeneous strain (piezoelectricity) and the inhomogeneous strain (flexoelectricity) has been concurrently generated at the free-standing region. Hence, the coexistence of the piezoelectric effect and the flexoelectric effect should be considered. Nonetheless, the flexoelectricity is an inherently size-dependent phenomenon. When the flake thickness decreases to the nanoscale dimension, the inhomogeneous strain dominantly overwhelms the homogeneous strain, which is simultaneously induced by tip-induced vertical shear strain (Supplementary Fig. 5). Additionally, cross-sectional field calculation[39] accurately correlates with the PFM image of $CuInP_2S_6$ nanowrinkle, $CuInP_2S_6$ nanogap (Fig. 2c), and free-standing flexoelectric simulation (see below). To further clarify the flexoelectric dominance in $CuInP_2S_6$ with nanoscale mechanical bending, flexoelectric field distribution in nanowrinkle has been observed in Supplementary Fig. 6a–c. As the nanowrinkle structure was generated without tip-induced vertical shear strain, internal polarization of the nanowrinkle has been dominantly generated from the local lattice bending and its corresponding flexoelectricity. As shown in Supplementary Fig. 6b, c, nanowrinkle PFM amplitude inversely correlates with the calculated surface potential of downward flexoelectricity[39], while the nanowrinkle PFM phase linearly corresponds with the surface potential of downward flexoelectricity[39]. Moreover, tip-induced strain engineering has been conducted within substrate suspension to further minimize the flexoelectric effects. Within substrate suspension, the applied tip force exhibits linear correlation with the distance (Supplementary Fig. 6d). Therefore, the applied tip force (480 nN, 600 nN, 720 nN) is insufficient to activate the lattice bending in the suspended substrate. In this case, plastic deformation can be completely hindered, which effectively minimizes the flexoelectric field effect. The hysteresis variation was also spatially mapped in the suspended substrate with each applied tip force, resulting in the absence of hysteresis variation of suspended $CuInP_2S_6$ (Supplementary Fig. 6e). Hence, the flexoelectric dominance was clearly observed in nanoscale geometry, constructed as the nanowrinkle structure and the suspended substrate[35,42,43].

## Programmable flexoelectric engineering for self-powered 2D memristor device

Unlike conventional geometric substrates, the axial nanogap structure permits a local free-standing state. To achieve spatial activation of ferro-ionic properties, vertical shear strain was applied to the free-standing states for efficient strain modulation, unlocking the deterministic 3D positioning of the ferro-ionic conduction. Figure 3a shows the schematic illustration of $CuInP_2S_6$ phase transition with spatial modulation of $Cu^+$ ion dynamics. According to vertical shear strain engineering, the $CuInP_2S_6$ phase transition was controlled in the mechanical bending geometry. When the vertical shear strain was applied to the free-standing area, a downward polarization enhancement locally emerged owing to the downward net polarization shift. The downward (upward) lattice bending exhibits the compensation (attenuation) direction of $Cu^+$ ion dynamics, which implies that the ferro-ionic conduction threshold voltage ($V_{th}$) can be reversibly controlled according to the lattice bending direction. As a result, the ferro-ionic conduction $V_{th}$ was decreased considerably under downward lattice bending, whereas upward lattice bending increased the ferro-ionic conduction $V_{th}$. To observe the initial stochastic domain distribution, upward bias poling of the $\alpha$-$In_2Se_3$/$CuInP_2S_6$ heterostructure was pre-characterized, which was statically clarified with the PFM domain pixel distribution (Supplementary Fig. 7). In the upward polarized $\alpha$-$In_2Se_3$ / $CuInP_2S_6$ heterostructure, $Cu^+$ and $Se^{2-}$ ions intrinsically generate a $Cu^+$-$Se^{2-}$ dipole at the interface, which facilitates the internal upward polarization. Based on the spatial $Cu^+$ ion transport, the non-deterministic spatial distribution is statically separated within the paraelectric and ferro-ionic phases, which is derived from the spatial dominance of $Cu^+$ ions and $Cu^-$ vacancies. The ion-dependent spatial dominance was correlated with the stochastic PFM pixel distribution. The PFM statical phase mapping indicates the dominant distribution of $Cu^+$ ions (−84.15°) and $Cu^-$ vacancies (89.39°), causing the discrete phase distribution. Correlatively, the PFM statical amplitude distribution was classified with the ferro-ionic $Cu^+$ ion dominant area (6.62 mV) and paraelectric $Cu^-$ vacancy dominant area (0.44 mV). In Fig. 3b, nanoscale $\alpha$-$In_2Se_3$ lattice bending was clearly observed in the cross-sectional TEM image and selected area electron diffraction (SAED) pattern, resulting in a downward net polarization shift and nanoscale flexoelectricity. Additionally, the force-distance curve and topography line profile indicate the accurate control of 3D tip positioning techniques in a three-dimensional scale (Supplementary Fig. 8a, b). Regarding the 3D tip positioning techniques, inhomogeneous mechanical strain can generate the programmable flexoelectric field, which implies the functionalized correlation of ferro-ionic current with respect to the scale of mechanical bending. To further clarify the flexoelectric enhancement of phase transition, PFM hysteresis was measured with flexoelectric energy conversion, as shown in Fig. 3c. Firstly, PFM amplitude hysteresis indicates the ferro-ionic phase transition at 5.43 V without the flexoelectric field (Supplementary Fig. 8c). In contrast, the flexoelectric field in the free-standing $\alpha$-$In_2Se_3$/ $CuInP_2S_6$ heterostructure reduces the ferro-ionic conduction $V_{th}$ from 5.36 V to 3.24 V, regarding the polymorphicity of $CuInP_2S_6$ in the consistent voltage range. According to the flexoelectric engineering, ferro-ionic phase is exhibited with a ferroelectric hysteresis in the range −10 to 3.24 V, resulting the memristive piezoelectric response in the range 6.75–10 V. To further clarify the reproducible paraelectric $V_{th}$ shift with various layer construction, exfoliated $\alpha$-$In_2Se_3$/$CuInP_2S_6$ flakes were selected for heterogeneous van der Waal layer counts. As a result, fully reversible paraelectric $V_{th}$ was measured with thickness-dependent layer dominance and flexoelectric energy storage. Under the flexoelectric field, paraelectric $V_{th}$ was clearly reduced with $\alpha$-$In_2Se_3$ dominant construction ($\alpha$-$In_2Se_3$ 120.70 nm / $CuInP_2S_6$ 115.65 nm), while the flexoelectric field facilitates the additional paraelectric conduction $V_{th}$ reduction. Nonetheless, $CuInP_2S_6$ dominant construction ($\alpha$-$In_2Se_3$ 33.36 nm and $CuInP_2S_6$ 180.37 nm) amplifies the paraelectric $V_{th}$, resulting in the reversible paraelectric $V_{th}$ reduction via downward polarization. (Supplementary Figs. 9 and 10). Additionally, the ferroelectric hysteresis loop was spatially mapped according to the axial nanogap

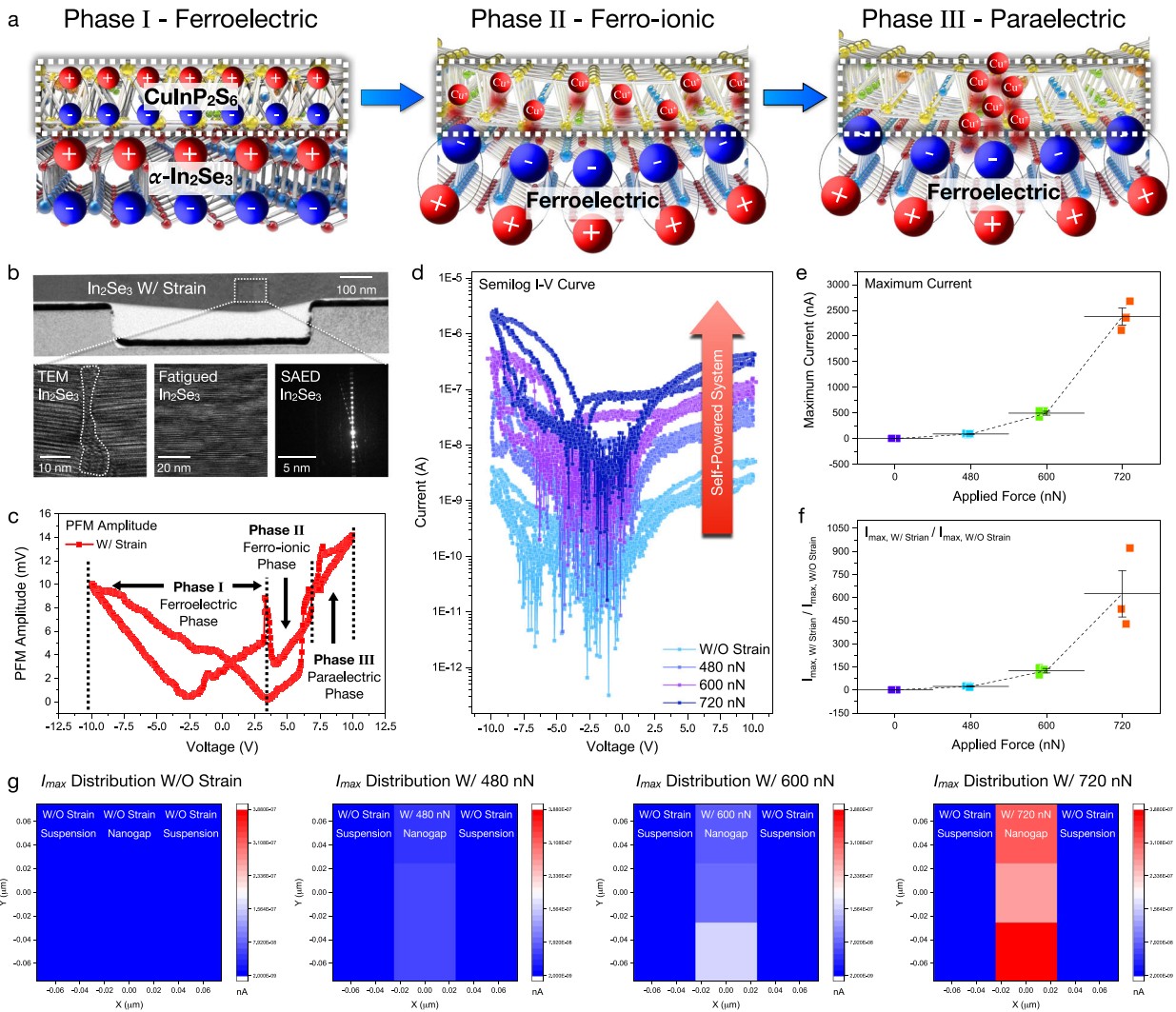

**Fig. 3 | Self-powered 2D ferro-ionic conduction via spatial flexoelectric nano-manipulation. a** Ferro-ionic phase transition mechanism in the 2D $\alpha$-In$_2$Se$_3$/CuInP$_2$S$_6$ heterostructure, which indicates the flexoelectric energy storage. **b** Cross-sectional TEM image and its corresponding SAED pattern of the bent In$_2$Se$_3$ lattice structure, which induces the nanoscale flexoelectricity. **c** PFM amplitude hysteresis behavior with nanoscale mechanical bending, indicating the conduction threshold reduction. Self-powered system in 2D ferro-ionic conduction, which was experimentally verified with **d** semi-log $I$–$V$ curve, **e** $I_{max}$ distribution, and **f** ratio of $I_{max, \text{with strain}}$ to $I_{max, \text{without strain}}$. **g** Spatial $I_{max}$ mapping in free-standing 2D $\alpha$-In$_2$Se$_3$/CuInP$_2$S$_6$ heterostructure, which enables the sub-50 nm flexoelectric manipulation.

topography to clarify the CuInP$_2$S$_6$ phase transition. Regardless of voltage range, the suspended region exhibited a homogeneous ferroelectric phase, whereas the nanogap region was observed to have a heterogeneous phase, exhibiting ferro-ionic and paraelectric phases within the respective ranges of −5 to 5 V and −10 to 10 V (Supplementary Fig. 11). To further validate the programmable Cu$^+$ ion transport, spatial $I_{max}$ mapping was conducted in free-standing 2D $\alpha$-In$_2$Se$_3$/CuInP$_2$S$_6$ heterostructure, as shown in Fig. 3g. Since the ferro-ionic conduction was spatially generated by nanoscale mechanical bending, robust ferro-ionic current amplification can be site-selectively observed. While the current has been measured as 4.13 nA without strain, the current level is significantly increased with tip-induced shear strain as 93.26 nA (480 nN), 497.67 nA (600 nN), and 2380.84 nA (720 nN). Within 720 nN of force, a $5.76 \times 10^2$-fold maximum current enhancement was clearly observed, corresponding to the ratio of 2380.84 nA (720 nN) to 4.13 nA (W/O strain). Additionally, $I_{max, \text{with strain}}$ / $I_{max, \text{without strain}}$ ratio was experimentally observed as 1 (W/O strain), 23.58 (480 nN), 125.46 (600 nN), and 625.41 (720 nN), which is attributed to the nanoscale flexoelectric energy storage. Moreover, spatial $I_{max}$ distribution was also mapped at the single $\alpha$-In$_2$Se$_3$ memristor and

single CuInP$_2$S$_6$ memristor, which correlatively indicates the programmable ferro-ionic conduction within sub-50 nm resolution (Supplementary Fig. 12). While the temperature-independency of ferro-ionic conduction is also significant for device applications, the temperature effects of ferro-ionic conduction have been further verified under 315 K, 330 K, 350 K, and 400 K conditions. According to the previous research[19], the CuInP$_2$S$_6$ phase has been classified as four heterogeneous states, i.e., the frozen-in polarization state, ferroelectric polarization state, Cu$^+$ ion hopping state, and conductive filament state. Because a temperature-dependent CuInP$_2$S$_6$ phase transition has been reported at $10^2$ MV/m and 300 K[19], the free-standing 2D nanogap device also exhibited temperature-dependent homogeneous ferro-ionic conductive behavior (Supplementary Fig. 13) under the 400 K condition. Moreover, the IV hysteresis behavior in −10 to 10 V sweep range is inhomogeneous, which is attributed to the ferro-ionic phase transition. While the −10 to 10 V range possesses the possibility of ferro-ionic phase transition within temperature variation, the −20 to 20 V sweep range clearly exceeds the ferro-ionic conduction $V_{th}$. Additionally, Cu filament formation was initiated under the 400 K condition, which is attributed to the paraelectric phase transition of CuInP$_2$S$_6$.

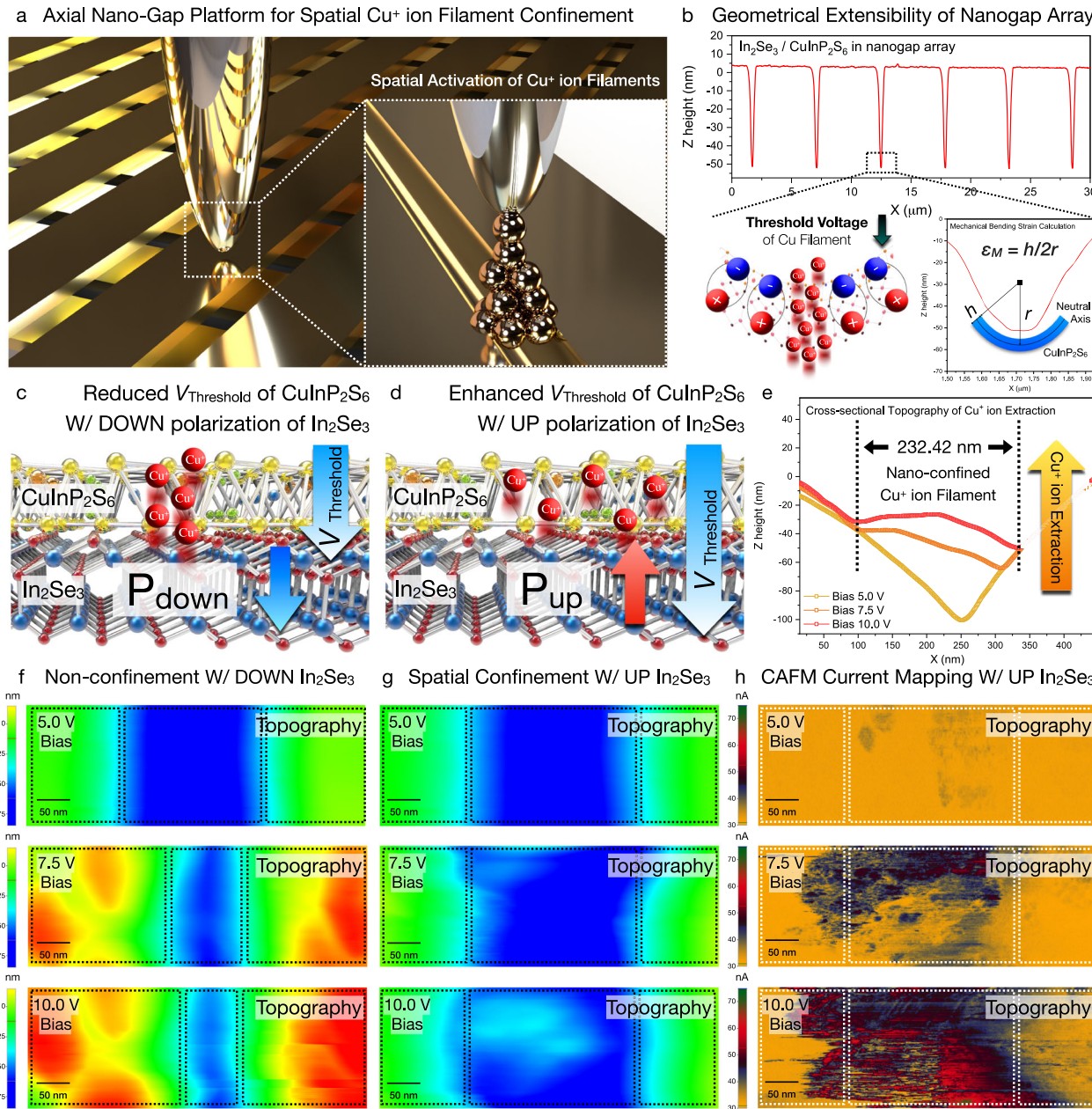

**Fig. 4 | Deterministic Cu⁺ ion transport via α-In₂Se₃ polarization and localized flexoelectric field. a** Schematic illustration of the free-standing 2D ferro-ionic memristor consisting of α-In₂Se₃/CuInP₂S₆ heterostructure. **b** Topographical line profile of mechanical bending curvature in the axial nanogap array. Reversible conduction threshold modulation through **c** downward polarization and **d** upward polarization of the bottom α-In₂Se₃. Conductive filaments were spatially observed within **e** homogeneous topographical variation with α-In₂Se₃ downward polarization and **f** inhomogeneous topographical variation with α-In₂Se₃ up polarization, which was correlatively verified with the **g** cross-sectional line profile and **h** CAFM current image.

## Spatial Cu⁺ ion modulation via reversible ferro-ionic conduction threshold shift

According to previous ferro-ionic studies, the flexoelectric engineering of CuInP₂S₆ has been actively reported[21,22,38,39]. These previous works are only available to control its ferroelectric domain switching with mechanical switching methods. While the mechanical bending, which was reported in previous works, was completely insufficient to activate the CuInP₂S₆ ferro-ionic conduction, the spatial modulation of conductive filaments has been completely blocked. To achieve spatial controllability of the conductive filaments, the concept of a free-standing 2D ferro-ionic memristor is presented in Fig. 4a. As shown in Fig. 4b, programmable flexoelectric engineering at the axial nanogap array was realized with an applied force of 600 nN, confirming the

robust reproducibility and reliability of tip-induced strain engineering. Regarding the geometric extensibility of the axial nanogap array, the spatial confinement of the conductive filaments can be homogeneously extended to a 6-inch wafer-scale axial nanogap structure (Supplementary Fig. 14). To unlock spatial ferro-ion controllability, an ultra-high bending curvature was selectively achieved in the axial nanogap array, allowing geometrical guidance of Cu⁺ ion transport. The nanoscale bending strain was measured to be 0.047 nm⁻¹ (480 nN), 0.064 nm⁻¹ (600 nN), and 0.071 nm⁻¹ (720 nN) in the freestanding area, corresponding to a $4.09 \times 10^2$-fold efficient bending strain and $1.63 \times 10^1$-fold efficient mechanical energy consumption relative to the previous values for the CuInP₂S₆ flexoelectric engineering at the nano-hole structure[38]. To reversibly control the CuInP₂S₆

ferro-ionic conduction $V_{th}$, the initial $\alpha$-In$_2$Se$_3$ polarization externally controlled the Cu$^+$ ion dynamics, as illustrated in Fig. 4c, d. While the $\alpha$-In$_2$Se$_3$ downward polarization decreased the initial ferro-ionic conduction $V_{th}$ with the homogeneous field direction, $\alpha$-In$_2$Se$_3$ upward polarization increased the initial ferro-ionic conduction $V_{th}$, owing to the heterogeneous field direction. Due to the reversible $\alpha$-In$_2$Se$_3$ polarization, Cu$^+$ ion dynamics were spatially controlled by ultra-high flexoelectric energy conversion efficiency. As tip-induced strain engineering only allows downward polarization, we spatially modulated the programmable shear strain 720 nN to achieve the paraelectric $V_{th}$ reduction. Within $\alpha$-In$_2$Se$_3$ downward polarization, homogeneous conductive filament growth and topographical variation (Fig. 4f) were clearly observed, which is attributed to the homogeneous paraelectric $V_{th}$ shift direction in both the free-standing and suspended areas. Nonetheless, the spatial confinement of the conductive filament was experimentally observed with $\alpha$-In$_2$Se$_3$ upward polarization, which can be attributed to the heterogeneous paraelectric $V_{th}$ shift between the free-standing and suspended states. While tip-induced strain engineering permitted localized paraelectric $V_{th}$ reduction with downward polarization, the paraelectric $V_{th}$ of the suspended area was increased within upward polarization. As the reversible paraelectric $V_{th}$ shift enabled the site-selective confinement of conductive filaments (Fig. 4g), Cu$^+$ ion extraction was spatially nanoconfined as 36.59 nm (sample bias 5 V), 44.93 nm (sample bias 7.5 V), and 58.98 nm (sample bias 10 V), which spatially correlated with the $I_{max}$ distribution as 32.81 nA (sample bias 5 V), 55.89 nA (sample bias 7.5 V), and 74.75 nA (sample bias 10 V). Moreover, nanoconfined topographical variation was further probed and clarified with cross-sectional topography line profile and conductive AFM (CAFM) current mapping, as shown in Fig. 4e, h.

## Theoretical validation of flexoelectric Cu$^+$ ion amplification in 2D ferro-ionic memristor

To further clarify the bidirectional threshold resistive switching in the free-standing 2D ferro-ionic memristor, As shown in Fig. 5a, the semi-log $I$–$V$ curve and corresponding energy band diagram were correlatively analyzed within 0 V → +10 V → −10 V → 0 V bias sweeping. The positive bias (0 V → 10 V) converts the low resistive state (LRS) to high resistive state (HRS) ([1] → [2]) at a ferro-ionic conduction $V_{th}$ as 6.75 V (Fig. 5b). As the Cu$^+$ ions drift upward in the positive-bias range, the inhomogeneous Cu$^+$ ion distribution facilitates an internal field consisting of Cu$^+$ ions and Cu$^-$ vacancies. Hence, the Cu$^+$ ions near the top Pt electrode surface operate as acceptors and the Cu$^-$ vacancies near the bottom Au electrode reversibly operate as donors, allowing the formation of $p$-$n$ homojunctions (Fig. 5c). Reversibly, negative bias sweep (0 V → −10 V) converts the HRS state to the LRS state ([3] → [4]) (Fig. 5d). While the Cu$^+$ ions drift downward in the negative bias range, they are redistributed to their original position in each CuInP$_2$S$_6$ lattice unit cell. As the negative bias sweep derives the inversion of the spatial Cu$^+$ ion distribution and $p$-$n$ homojunction direction, bidirectional threshold resistive switching behavior is clearly observed (Fig. 5e). Additionally, single 2D memristor performance was characterized with full $I$–$V$ curve measurements, resulting the conventional memristive behavior unlike 2D $\alpha$-In$_2$Se$_3$ / CuInP$_2$S$_6$ heterostructure (Supplementary Fig. 15). Furthermore, a numerical simulation of ferroelectric polarization was performed to theoretically validate the observations of the flexoelectric effects. The total Gibbs free energy of the system can be calculated within the applied field configuration, as

$$G_{tot} = \int_V (G_{bulk} + G_{grad} + G_{plas} + G_{elec} + G_{flexo})\, dV \tag{1}$$

In Eq. (1), V denotes the system volume, $G_{bulk}$ indicates the Gibbs free energy, $G_{grad}$, $G_{elec}$, $G_{plas}$, and $G_{flexo}$ imply the Gibbs free energy of the gradient, plasticity, electrostatic, and flexoelectricity, respectively. As

shown in Fig. 5f–j, a free-standing 2D ferroelectric heterostructure was modeled with consideration of piezoelectric field and flexoelectric field to accurately quantify the flexoelectric energy conversion, which accurately matches with previous flexoelectric simulation results[21,22,39,44]. Within the actual bending curvature values, the polarization was numerically calculated as 0.010 $\mu$C/m$^2$ (480 nN), 0.013 $\mu$C/m$^2$ (600 nN), and 0.015 $\mu$C/m$^2$ (720 nN), indicating an analogous correlation with the experimental results. Additionally, strain-polarization simulations predicted strain tensor values as 0.86 N/m$^2$ (480 nN), 1.10 N/m$^2$ (600 nN), and 1.28 N/m$^2$ (720 nN), which originated from the flexoelectric energy conversion and field-dependent piezoelectric response (Supplementary Fig. 16). To further quantitatively estimate the mechanical bending dimensionality, the flexoelectric equation was derived.

$$G_{flexo} = \frac{1}{2} G_{ijkl} \left( \frac{\partial P_k}{\partial x_l} e_{ij} - \frac{\partial e_{ij}}{\partial x_l} P_k \right) \tag{2}$$

As flexoelectricity indicates the inherent scale dependence, flexoelectricity dominance emerges at nanoscale material dimensions rather than microscale material dimensions. According to Eq. (1), the strain gradients exhibited amplified polarization during downward bending. Moreover, the local polarization redistribution and amplification were theoretically supported by performing Gibbs free energy modeling. The simulation revealed that, as the bending curvature increases, the amplitude of the flexoelectric field proportionally increases, which can be further clarified by the previous nanogap research including a microscale mechanical bending that also yielded a significantly lower bending curvature than the nanoscale mechanical bending[44]. Furthermore, the vertical stacked 2D $\alpha$-In$_2$Se$_3$/CuInP$_2$S$_6$ enhances the total ferroelectric polarization by Cu$^+$-Se$^{2-}$ dipole interaction, leading to larger ferroelectric hysteresis behavior. To precisely evaluate the ferroelectric hysteresis enlargement within the 2D ferroelectric vertical stacking, Gibbs free energy modeling of the $\alpha$-In$_2$Se$_3$/CuInP$_2$S$_6$ heterostructure was theoretically considered within the structural configuration as in Eq. (3)[45].

$$G_{tot} = G_{f1} t_{f1} + G_{f2} t_{f2} = (\alpha d_{f1} + x t_{f2}) P_s^2 + (\beta t_{f1} + y d_{f2}) P_s^4 - V P_s \tag{3}$$

In Eq. (3), $G_{f1}$ is the Gibbs free energy of the ferroelectric semiconductor $\alpha$-In$_2$Se$_3$, and $G_{f2}$ is the Gibbs free energy of the ferroelectric dielectric CuInP$_2$S$_6$. A, $\beta$ imply the Landau coefficient of $G_{f1}$, and (x, y) represents the Landau coefficient of $G_{f2}$. Also, the flake thicknesses are indicated as $d_{f1}$ and $d_{f2}$, while $P_s$ represents the spontaneous polarization. Lastly, V ($V = V_{f1} + V_{f2}$) denotes the voltage of $\alpha$-In$_2$Se$_3$/CuInP$_2$S$_6$. To quantitatively clarify the polarization in ferroelectric heterostructures, Gibbs free energy modeling can be modified as a function of $P_s$ and $V_{f1}$. As the $d_{f1}$, $d_{f2}$, x, and y values are consistent without the external conditions, $V_{f1}$ can be expressed as the sum of the linear function and cubic function of $P_s$.

$$V_{f2} = -2x d_{f2} P_s - 4y d_{f2} P_s^3 \tag{4}$$

From the relative combination of Eqs. (1) and (3), the Gibbs free energy can be simultaneously considered in terms of both the field and structural configuration. The Gibbs free energy has two stable $P_s$ states, where the $G_{tot}$ derivative is calculated as 0. In the initial states, one distinctive $P_s$ state can be clearly observed, which is correlated with Cu$^+$-Se$^{2-}$ dipole formation. When $V_{f2}$ gradually increases, distinctive $P_s$ states occur sequentially as functions of the graph intersection counts. In the overlapping negative capacitance range, graph intersection counts gradually increase from one distinctive $P_s$ state to three distinctive $P_s$ states, similar to the reversible fashion. After negative capacitance interference, one distinctive $P_s$ state can be derived and

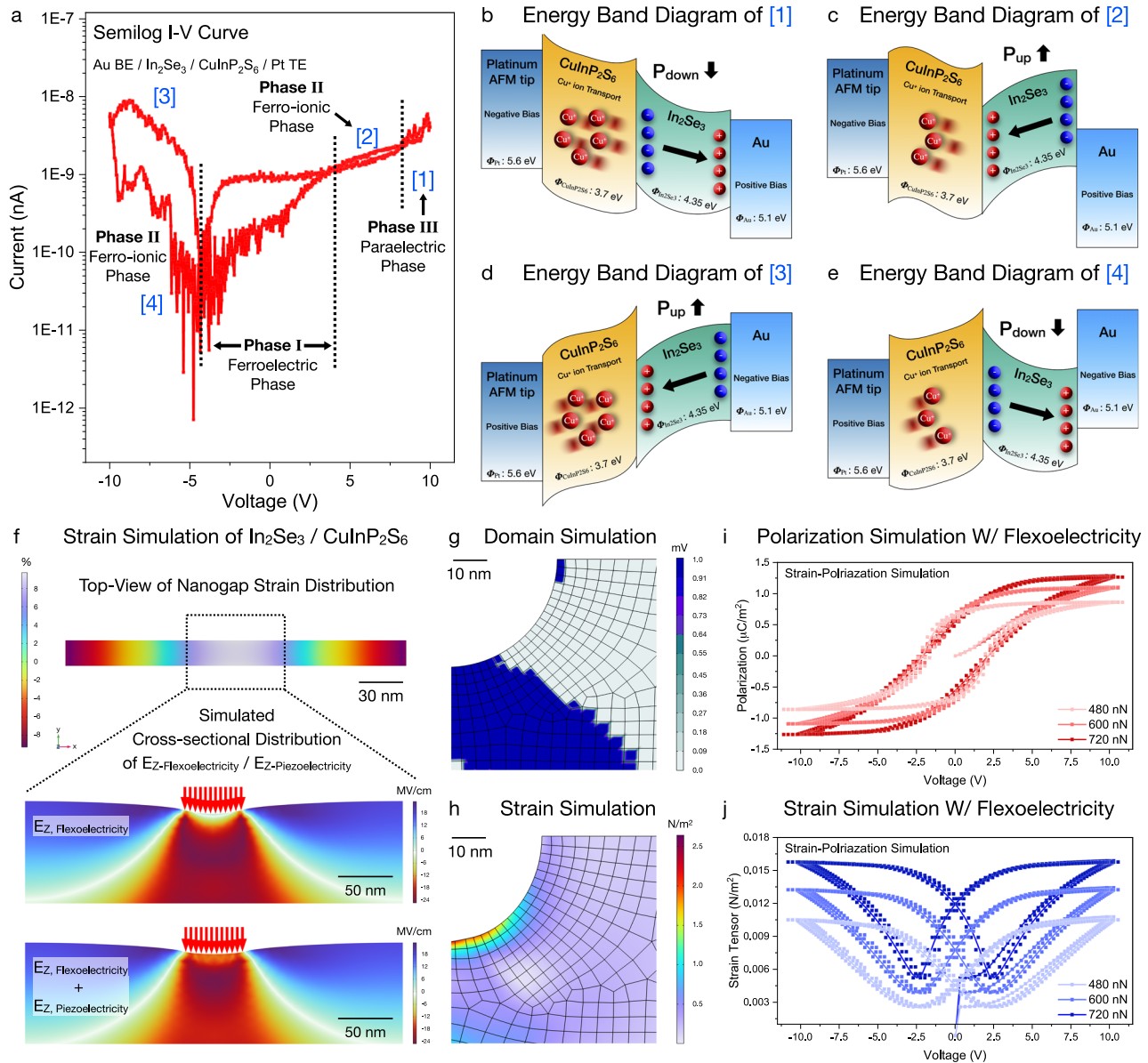

**Fig. 5 | Theoretical validation of Cu⁺ ion dynamics within nanoscale flexo-electric engineering. a** Full $I–V$ curve of a single 2D ferro-ionic memristor and **b** its corresponding energy band diagram of **b** upward $\alpha$-In₂Se₃ / downward CuInP₂S₆, **c** downward $\alpha$-In₂Se₃ / downward CuInP₂S₆, **d** downward $\alpha$-In₂Se₃ / upward CuInP₂S₆, and **e** upward $\alpha$-In₂Se₃ / upward CuInP₂S₆. **f** Strain calculation of the free-standing 2D ferroelectric heterostructures with consideration of piezoelectric field and flexoelectric field, exhibiting the cross-sectional distribution of the **g** domain and **h** strain. Theoretical evaluation of flexoelectric energy storage, resulting in the **i** bias-polarization hysteresis and **j** strain-polarization hysteresis.

maintained in a high $V_{f2}$ range. As the flexoelectric field enables hopping of the overlapped negative capacitance range, the numerical validation is deterministically correlated with the experimental $P_s$ state counts within a homogeneous $V_{f2}$ range, which theoretically supports hysteresis amplification through flexoelectricity and vertically stacked ferroelectric materials.

## Discussion

In this study, we presented a free-standing axial nanogap platform for deterministic ferro-ionic modulation through nanoscale flexoelectric engineering. To spatially modulate the Cu⁺ ion, an ultra-high vertical shear strain was site-selectively facilitated in the free-standing area to activate the ferro-ionic conduction, which allowed local positioning of the CuInP₂S₆ phase transition. Moreover, we experimentally demonstrated the concept of a single 2D ferro-ionic memristor with nanoscale flexoelectric engineering. As the local flexoelectric energy storage

results in a fully reversible paraelectric $V_{th}$ shift, a $6.25 \times 10^2$-fold increased $I_{max, with strain}$ / $I_{max, without strain}$ ratio was experimentally observed owing to the vertical shear strain 720 nN, which is theoretically supported by the 3D flexoelectric simulation. Additionally, a topographical Cu⁺ ion extraction was locally confined within 232.42 nm width and 58.98 nm height, while the upward polarized bottom In₂Se₃ suppresses the undesirable ionic conduction in the suspended junction area. Therefore, we envision that our free-standing 2D ferro-ionic memristor provides an extendable geometric solution for ultra-efficient self-powered system and reliable neuromorphic computing systems.

## Methods

### Axial nanogap fabrication

The axial nanogap substrate was fabricated at the National Nano Fab Center (NNFC, etching team, Republic of Korea) and sequenced as a

front-end-of-line poly Si etch, normal photoresist (PR) strip, followed by post-cleaning and PR coating. After fabrication, the nanogap substrate was rinsed in acetone solution for 3 min at 500 RPM to delaminate the passivation PR layer. After the PR removal, residual organic contaminants were removed using deionized water. A Ti adhesion layer (2 nm thickness) and Au film (18 nm thickness) were deposited on the nanogap structure. The deposition conditions were 0.2 A/s (deposition rate), $8 \times 10^{-7}$ torr (deposition pressure), room temperature (deposition temperature), and 5 RPM (rotation rate) through the electron evaporation system (KVE-E2000L, KOREA VACUUM TECH., Republic of Korea).

### Mechanical exfoliation and transfer of 2D ferroelectric materials

Prior to mechanical exfoliation and dry transfer, a polydimethylsiloxane stamp was attached to the cover glass. As the 2D ferroelectric materials ($\alpha$-In$_2$Se$_3$/CuInP$_2$S$_6$) were mechanically exfoliated from the bulk crystals (HQ Graphene, Netherlands) onto polydimethylsiloxane stamps, $\alpha$-In$_2$Se$_3$ was transferred to the nanogap. Following $\alpha$-In$_2$Se$_3$ dry transfer, CuInP$_2$S$_6$ was arbitrarily transferred above the $\alpha$-In$_2$Se$_3$ position by applying the 70 °C transfer condition, which determined the local free-standing states. Alternatively, the nanowrinkle structure was naturally generated during the dry transfer process (Supplementary Fig. 17).

### Atomic force microscopy

AFM (NX-10, Park Systems, Republic of Korea) measurements were conducted using an ElectriMulti75-G cantilever, with a silver paste electrode selectively deposited at the edge of the sample to induce vertical charge transfer. The ElectriMulti75-G cantilever was calibrated with a tip radius (25 nm), length (225 μm), height (17 μm), width (28 μm), and spring constant (3.3 N/m), which resulted in 320.8 kHz contact resonance frequency. During the tip-induced strain engineering, the inhomogeneous shear strain was vertically controlled with a gradual force limitation (480, 600, and 720 nN), hold time (50 s), and speed (0.5 μm/s). After the tip-induced strain engineering, AFM topography was measured with ElectriMulti75-G cantilever, contact mode, scan rate 0.7 Hz, and set point 0.3 V.

### Material characterization

To pre-characterize the 2D ferroelectric heterostructure, X-ray photoelectron spectroscopy (XPS) measurements (NEXSA, Thermo Fisher Scientific, USA) were performed with an X-ray spot size of 400 μm. XPS Peak deconvolution was performed on the Cu 2$p$, In 3$d$, P 2$p$, S 2$p$, and Se 3$d$ spectra. The XPS profiles were aligned using the C 1 s peak at a binding energy of 285 eV. After the measurements, the data were calibrated using CASAXPS software (version 8.1). The transferred flakes were characterized using optical microscopy (U-MSSP4, Olympus, Japan) and FE-SEM (S-4800, Hitachi, Japan). A focused ion beam instrument (NX2000, Hitachi Ltd., Japan) was used to prepare the cross-sectional TEM specimens using a Ga$^+$ ion beam from 30 to 5 keV, and a lift-off process was conducted to etch the specimens. TEM (JEM-2100F, JEOL, Japan) and X-ray diffraction (Empyrean, Malvern PANalytical, United Kingdom) were used to observe the lattice structure, EDS mapping, and SAED pattern of the layered structures of $\alpha$-In$_2$Se$_3$ and CuInP$_2$S$_6$ at the atomic scale.

### 3D flexoelectricity simulations

To theoretically quantify the nanoscale mechanical bending, a 2D $\alpha$-In$_2$Se$_3$/CuInP$_2$S$_6$ heterostructure was modeled using the commercially available 3D simulation program COMSOL Multiphysics 6.1. The experimental material characteristics, including Young's modulus, Poisson's ratio, domain wall density, and polarization reversibility, were extracted from previous studies[38,46–48] and input to the 3D ferroelectric simulation model. The In$_2$Se$_3$/CuInP$_2$S$_6$ heterostructure was numerically modeled with experimental layer dimensionality at room temperature, a free-standing state, and a lateral width of 500 nm, which enabled a similar mechanical bending with experimental values. Under these conditions, the strain-polarization coupling was calculated as a function of the mechanical bending curvature (420 nN, 600 nN, 720 nN), which was experimentally obtained for a free-standing 2D ferroelectric heterostructure.

### Reporting summary

Further information on research design is available in the Nature Portfolio Reporting Summary linked to this article.

## Data availability

All other data that support this study are available from the corresponding authors upon reasonable request. Source data are provided with this paper.

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

## Acknowledgements

This research was supported by the Basic Science Research Program of the National Research Foundation of Korea (NRF), funded by the Ministry of Education (No. 2022R1A3B1078163). This research was supported by the MOTIE (Ministry of Trade, Industry, & Energy) (1415180243) and Korea Semiconductor Research Consortium (KSRC) (20020410) support program for the development of future semi-conductor devices. This work was supported by the Institute for Basic Science (IBS-R027-D1). This work was supported by the Technology Innovation Program (Public-private joint investment semiconductor R&D program(K-CHIPS) to foster high-quality human resources) (RS-2023-00235484, "Development of High Quality MX2 Materials and Processes through In-situ Defect Analysis") funded By the Ministry of Trade, Industry & Energy (MOTIE, Korea) (1415187770). The authors thank H. J. Cho (National Nano Fab Center) for the nanogap substrate fabrication and the corresponding discussion.

## Author contributions

J.L. and G.W. contributed equally to this study. J.L. and G.W. prepared the samples and performed the main experiments and simulations. J.C. performed the single 2D ferro-ionic memristor device fabrication. S.S., H.Shin, and H.Seok were involved in the technical discussions on con-ductive filament formation. M.K., E.K., H.Seok, and Z.W. conducted the analytical experiments, including the TEM, EDS, XPS, XRD, and SEM measurements. B.K. and W.J. technically discussed the structural properties of nanogap structure with J.L. J.L., G.W., J.C., S.S., H.Seok, B.K., and W.J. analyzed and discussed the experimental results. J.L. wrote the manuscript with contributions from all authors. T.Kim designed and supervised the study. All authors have read and approved the final version of this manuscript.

## Competing interests

The authors declare no competing interests.
