## [Peer Review File · Nature Communications]

Free-standing two-dimensional ferro-ionic memristorREVIEWER COMMENTS

Reviewer #1 (Remarks to the Author):

Jinhyoung et al. investigated a nanogap array for nano-confined conductive filament growth in a free-standing 2D ferro-ionic memristor. AFM tip is used to press the 2D CuInP2S6/In2Se3 heterostructure and introduce strain. The strain is tunable by applying different forces on the tip. Ferro-ionic and paraelectric states can be tuned through electrical gating. While the concept contributes to the field, some technical problems need to be addressed, and the manuscript shall be revised to increase clarity and make the contributions and limitations clear. I would consider for publication only if the following points are addressed appropriately.

1. The authors claimed a 576-fold maximum current enhancement. But this number only shows up in the abstract and introduction. What does it mean? What is the baseline? There is no explanation in the main text.
2. The authors claimed ultra-high strain gradient and ultra-high flexoelectric energy conversion efficiency but specific values are missing to support the term "ultra-high".
3. On page 5 line 109, Fig. 1c is not TEM and EDS mapping. It should be Fig. 1b. The "real" Fig. 1c Memristor Cross-bar is not mentioned in the manuscript.
4. In the EDS mapping, the thickness of In and Se are comparable to the scale bar (250 nm), and Cu and P are thicker than the scale bar (250 nm). From the figure, I estimate In2Se3 to be around 250 nm and CIPS around 300nm. Why it is stated as " α -In2Se3 (33.36 nm), CuInP2S6 (180.37 nm)" on page 5 line 110?
5. How is the profile measured in Fig. 1e? It is just scanning the sample while applying force on the tip? If so, the sample profile may not be in equilibrium during the scanning, and scanning forward and backward may give different profiles. What is the scanning speed? Why reference at 480 nN instead of 0 nN? Why is the curvature linearly correlated to the force?
6. In Fig. 1f, the colors of 480nN and 600nN are indistinguishable. I would kindly suggest using a higher-contrast color map.
7. How is the nano-wrinkle structure fabricated in Fig. 2? Is it just finding some random wrinkle on the sample?
8. Some confusing statements need to be refined. For example:
 - o "Nonetheless, previous research was completely limited due to the geometrical substrate suspension and anisotropic strain gradient, insufficient strain gradient, since the axial nanogap platform (Fig. 1d) derives the local free-standing state and an efficient strain gradient, unlike the conventional geometrical substrate".
 - o "To further clarify the localized electron drift, the heterogeneous electron distribution included the suspended area, compressive strain area, and tensile strain area."
9. The authors state that "According to the geometric extendable axial nanogap array, wafer-

scale spatial confinement of the conductive filaments was achieved, as shown in Fig. 4b”
However, only a 30 um profile was shown in Fig. 4b, not wafer-scale. And what is the force for Fig. 4b?

10. The scale bar is missing in Supplementary Fig. 1a, almost invisible in Supplementary Fig. 3 b, c, e, f

11. There is no definition of CuInP2S6 as CIPS. But many figures labeled CIPS.

12. The authors mentioned that CuInP2S6 was aligned and transferred to In2Se3. Are two straight edges from the two flakes aligned? Or just stack CIPS on top of In2Se3 with arbitrary orientation?

Reviewer #2 (Remarks to the Author):

The authors report a proof-of-concept memristor based on CuInP2S6/In2Se3 which is programmed using a PFM tip. I found the article to be poorly written and rather difficult to follow. More importantly, I believe the authors give a physical explanation of the mechanisms at play that is not correct, since the flexoelectric coupling seems to have been misunderstood, and the piezoelectric effect (which may be even larger than the flexoelectric effect) is completely overlooked. Also, the double phase transition undergone by the CuInP2S6 (CIPS) layer which is partly responsible for the programmability is not sufficiently backed up by the measurements (or the model) presented by the authors. My recommendation is therefore to not publish the manuscript in Nature Communications.

I now detail my main concerns:

(1) Flexoelectricity seems to have been misunderstood by the authors, hence the physical mechanism used to explain their observations seems wrong. A compressive or a tensile strain on their own do not induce a change in polarization via a flexoelectric coupling, but rather via a piezoelectric coupling. It is only a strain gradient that induces a change in the polarization of a flexoelectric origin. Instead, the authors often refer to the flexoelectric origin of a polarization in regions of compressive or tensile strain, which therefore leads to wrong statements. The regions under homogeneous strain, like for instance those indicated in Fig. 2a as “downward flexoelectricity” and in 2b as “upward flexoelectricity”, which are around the inflection point of the indented membrane (and hence with a very small or null average strain gradient), should show no net flexoelectricity. (The terms upward and downward flexoelectricity can be misleading and are not well defined). Also, the terms tensile and compressive strain are used for the central region around the PFM tip in the nano-wrinkle and nano-gap geometries, respectively. However, the region around the gap is in both cases under in-plane tensile strain, since the structure is most likely stretched around the central point. Therefore, the whole discussion is very misleading, and some conclusions made on the basis of the sign of the strain in the central region are necessarily wrong, or at least wrongly stated.

(2) Equally importantly: CIPS is piezoelectric, and hence the piezoelectric response should play an important role in the picture – it is in fact the lowest order non-zero term in the polarization-strain coupling for this material and should therefore be considered. Still, the authors have essentially ignored this possibility. I believe the authors should consider piezoelectric effects when explaining their findings or explain why they can be ruled out.

(3) The phase transitions shown in Fig. 5 (from ferroelectric to ferroionic to paraelectric) seem to be deduced only from the I-V curve shown in Fig. 5a. However, other mechanisms could be at play here (rupture of the sample due to indentation, deposition of atoms from the tip onto the sample, ...). I believe that the authors should back this statement with further measurements – for example, if the ferroelectric phase is truly ferroelectric, they could measure ferroelectric hysteresis loops (P-E) with the PFM. And when it becomes paraelectric, it should show a characteristic paraelectric behavior on a P-E loop.

(4) The geometry of the whole device is not clear to me from Fig.1. In panel 1c, where does the tip make the indentation? It seems like there is no place where the CIPS layer is accessible.

Some (relatively) minor concerns:

-Shear strain and bending curvature values are given in units of nN, but those are units of force and hence cannot be the right units for either of those quantities.

-I could not understand several sentences in the manuscript, where the authors seem to be misunderstanding concepts. These are the most obscure to me:

- "The local flexoelectric energy storage results in 5.76×10^2 -fold increased maximum current was observed within vertical shear strain 720 nN"
- "the flexoelectric energy conversion induced an internal electrical field"
- "the heterogeneous electron distribution included the suspended area"
- "flexoelectric energy storage in free-standing α -In₂Se₃/CuInP₂S₆ heterostructure leads to 1.3272 V ferro-ionic conduction shift"

• "As flexoelectricity indicates the inherent scale dependence, flexoelectricity dominance emerges at nanoscale material dimensions rather than microscale material dimensions"

Some other sentences require further explanations or context, which is not provided:

- "According to previous ferro-ionic research, the spatial modulation of conductive filaments was completely blocked". This statement asks for further elaboration, plus a reference to the previous research mentioned.
- "According to equation (1), it is evident that both the microscale and nanoscale strain gradients exhibit flexoelectric polarization amplification within compressive strain, while the localized polarization distribution and field intensity of the microscale strain gradient is significantly lower than the nanoscale strain gradient". No local considerations can be derived -or at least they are not evident at all- from Eq.1, which gives a macroscopic Gibbs free energy. Also, compressive strain itself does not induce a flexoelectric polarization (rather a change in strain does). Finally no distinction between microscale and nanoscale behavior can be deduced from Eq. (1).

-The figures for the (thicknesses?) of the layers conforming the system are given in nm with two and sometimes three significant decimal numbers, implying sub-angstrom resolution, but I believe the methods used do not have that actual resolution. Many other figures in the manuscript also imply very high resolutions (voltages, angles), but I am no expert in typical PFM output signals and cannot judge the validity of those implied precisions.

-Fig. 5f is missing the legend for the color bar. Also, I believe it would be more illustrative to show a map of the strain gradient in the sample, rather than the actual strain. This would show where flexoelectricity should induce larger changes in the polarization.

-I could not understand the following sentence: “the force-distance curve and current-distance curve indicates the coupling of the vertical shear strain and strain gradient in the free-standing area, using programmable flexoelectric energy conversion”. The indentation with the tip induces vertical (shear?) strain locally, which results in a strain gradient between the region around the tip and the regions away from it. I do not understand what coupling the authors are thinking of and what role does the programmable flexoelectric energy conversion play here.

-The abstract is hard to understand because it lacks contextualization. For example, the “external conditions” the authors refer are only later explained in the text, but in the abstract they seem something special and unexplained.

-I do not understand what are the “heterogeneous” and “homogeneous” field directions (page 7, towards the end).

Reviewer #3 (Remarks to the Author):

This article reported a programmable flexoelectric energy storage platform in free-standing 2D ferro-ionic memristor. This universal nanogap platform can provide the extendable geometric solution for nanoscale flexoelectric energy conversion efficiency and reliable neuromorphic device. Although CuInP_2S_6 has been reported many times for the neuromorphic memristor, this manuscript offered a new memristor via a novel approach, the results are interesting, the measurement data are solid. I recommend publication after minor revision.

Temperature-independence ferro-ionic conduction behavior is very important for the device application. therefore, it would be better if authors can offer some Temperature-independent conduction or current behavior.

Response to the Nature Communications Reviewers' Comments

Dear reviewers,

We the authors deeply thank the reviewers for their thorough reading of our manuscript and the constructive comments. With regard to the reviewers' concerns raised, we have revised the manuscript and supporting information based on the reviewers' comments.

The reviewers' comments appear in black, and the authors' responses in blue. In the revised manuscript, the changes with respect to the previous version are highlighted.

Sincerely,

Taesung Kim

=====

Point-by-point Response

=====

Reviewer (#1)'s COMMENTS:

Jinhyoung et al. investigated a nanogap array for nano-confined conductive filament growth in a free-standing 2D ferro-ionic memristor. AFM tip is used to press the 2D $\text{CuInP}_2\text{S}_6/\text{In}_2\text{Se}_3$ heterostructure and introduce strain. The strain is tunable by applying different forces on the tip. Ferro-ionic and paraelectric states can be tuned through electrical gating. While the concept contributes to the field, some technical problems need to be addressed, and the manuscript shall be revised to increase clarity and make the contributions and limitations clear. I would consider for publication only if the following points are addressed appropriately.

Response:

We thank the reviewer for the positive evaluation with the constructive comments to improve the completeness of our work. Based on the raised concerns, we have performed

additional experiments and point-by-point responses with corresponding revisions to our revised manuscript.

Review #1_Q1) The authors claimed a 576-fold maximum current enhancement. But this number only shows up in the abstract and introduction. What does it mean? What is the baseline? There is no explanation in the main text.

Response:

Thank you very much for pointing out the missing explanations in the main text. The mentioned maximum current enhancement (576-fold) indicates the level of current amplification according to the presence of strain, and the value corresponds to the ratio of 2380.84 nA (720 nN) to 4.13 nA (W/O strain). We thank the reviewer for pointing this out and have revised the sentence as follows:

[Revised Text] Main text (Page 10):

Since the ferro-ionic conduction was spatially confined owing to the nanoscale shear strain, flexoelectric energy storage led to robust ferro-ionic current amplification, with values measured to be 4.13 nA (W/O strain), 93.26 nA (480 nN), 497.67 nA (600 nN), and 2380.84 nA (720 nN). Within 720 nN of force, a 5.76×10^2 -fold maximum current enhancement was clearly observed, corresponding to the ratio of 2380.84 nA (720 nN) to 4.13 nA (W/O strain).

Review #1_Q2) The authors claimed ultra-high strain gradient and ultra-high flexoelectric energy conversion efficiency but specific values are missing to support the term "ultra-high".

Response:

We apologize for the insufficient explanation of the term "ultra-high" in ultra-high strain gradient and ultra-high flexoelectric energy conversion. Actually, we have already

mentioned relative efficiency (40.967-fold efficiency) to support the term “ultra-high strain gradient” on page 10, line 217 (original article file). To further clarify the relative efficiency of the axial nanogap platform, we have selected the two significant references about CuInP_2S_6 flexoelectricity to relatively compare within the axial nanogap structure. Recently, the active flexoelectric control of CuInP_2S_6 was previously reported within a suspended substrate (Jiang, Xingan, et al. *Nature communications* **13.1** (2022)) and nano-hole structure [Lun, Yingzhuo, et al. *Advanced Materials* **35.36** (2023)]. In CuInP_2S_6 suspended substrate, ferro-ionic current was amplified for 4.36-fold within voltage sweep range correction and gradually-increased vertical shear strain from 0.35 μN to 4.20 μN . Conversely, the axial nanogap platform exhibits the 5.76×10^2 -fold current amplification from 480 nN to 720 nN force, which robustly supports the term “ultra-high”. At the nano-hole structure, CuInP_2S_6 was mechanically deformed within 18.16 μN indentation, which is 16.3-fold larger applied force than the elastic limit of the axial nanogap structure (480 nN), which also confirms the term “ultra-high”. We thank the reviewer for pointing this out and have revised the original text with the additional explanation as follows:

[Revised Text] Main text (Page 12):

The nanoscale bending strain was measured to be 0.047 nm^{-1} (480 nN), 0.064 nm^{-1} (600 nN), and 0.071 nm^{-1} (720 nN) in the free-standing area, corresponding to a 4.09×10^2 -fold efficient bending strain and 1.63×10^1 -fold efficient mechanical energy consumption relative to the previous values for the CuInP_2S_6 flexoelectric engineering at the nano-hole structure [Lun, Yingzhuo, et al. *Advanced Materials* **35.36** (2023)].

Review #1_Q3) On page 5 line 109, Fig. 1c is not TEM and EDS mapping. It should be Fig. 1b. The “real” Fig. 1c Memristor Cross-bar is not mentioned in the manuscript.

Response:

We apologize for our mistake. We thank the reviewer for pointing this out and have corrected the sentence as follows:

[Revised Text] Main text (Page 5):

As shown in Fig. 1b, the free-standing 2D ferro-ionic memristor consisted of an Au/ α -In₂Se₃/CuInP₂S₆/Pt structure, which was fabricated with the dry transfer method at the axial nanogap structure (Supplementary Fig. 1 and 2). Cross-sectional transmission electron microscopy (TEM) and corresponding energy dispersive spectroscopy (EDS) element mapping (Fig. 1b) were conducted to visualize the device structure of the 2D ferro-ionic memristor, which constructed with an Au bottom electrode (20 nm), α -In₂Se₃ (125.37 nm), CuInP₂S₆ (197.91 nm), and Pt tip as the top electrode. Additionally, device construction of 2D ferro-ionic memristor was exhibited with 3D schematic illustration for structural clarification and 3D stacking order (Fig. 1c).

Review #1_Q4) In the EDS mapping, the thickness of In and Se are comparable to the scale bar (250 nm), and Cu and P are thicker than the scale bar (250 nm). From the figure, I estimate In₂Se₃ to be around 250 nm and CIPS around 300nm. Why it is stated as " α -In₂Se₃ (33.36 nm), CuInP₂S₆ (180.37 nm)" on page 5 line 110?

Response:

We thank the reviewer for pointing out the mismatch between the TEM cross-sectional image and its corresponding device structure. Since a total of four free-standing ferro-ionic memristors were fabricated for AFM spectroscopy analysis, consisted of [1] CAFM device (Fig. 1b), [2] reference PFM device (Supplementary Fig. 10b), [3] α -In₂Se₃ dominant PFM device (Supplementary Fig. 10a), and [4] CuInP₂S₆ dominant PFM device (Supplementary Fig. 10c). As the [1] CAFM measurements clearly represent the originality of this work, we have revised the original text and Fig. 1b with [1] CAFM device structure (Au bottom

electrode (20 nm), α - In_2Se_3 (125.37 nm), CuInP_2S_6 (197.91 nm), and Pt tip as top electrode). We thank the reviewer for pointing out this discrepancy and have revised the sentence as follows:

[Revised Text] Supplementary Information (Page 2):

Supplementary Fig. 1 has been edited as follows.

Supplementary Fig. 1. Structural characterization of axial nanogap structure. (a) Optical image of the axial nanogap structure, consisting of 500 nm width and 150 nm depth. (b) SEM image of fabricated axial nanogap array. (c) Optical image of free-standing 2D α - In_2Se_3 / CuInP_2S_6 heterojunction on the axial nanogap structure and (d) its corresponding cross-sectional TEM image of free-standing 2D α - In_2Se_3 / CuInP_2S_6 heterojunction. (e) Line profile and (f) its corresponding CAFM IV curve at the free-

standing 2D ferroelectric heterojunction with various applied force, enabling the comparison with free-standing states (0 nN).

[Revised Text] Main text (Page 5):

Cross-sectional transmission electron microscopy (TEM) and corresponding energy dispersive spectroscopy (EDS) element mapping (Fig. 1b) were conducted to visualize the device structure of the 2D ferro-ionic memristor, which constructed with an Au bottom electrode (20 nm), α -In₂Se₃ (125.37 nm), CuInP₂S₆ (197.91 nm), and Pt tip as the top electrode.

Review #1_Q5) How is the profile measured in Fig. 1e? It is just scanning the sample while applying force on the tip? If so, the sample profile may not be in equilibrium during the scanning, and scanning forward and backward may give different profiles. What is the scanning speed? Why reference at 480 nN instead of 0 nN? Why is the curvature linearly correlated to the force?

Response:

We agree with the reviewer that we should provide more information about AFM metrology and strain engineering protocol. Firstly, nanoscale bending curvature (Figure. 1e) was independently manipulated at the free-standing gap region through the AFM nano-indentation. After the nano-indentation with AFM tip, 3D topography was measured with conventional AFM scanning with ElectriMulti75-G cantilever, contact mode, scan rate 0.7 Hz, and set point 0.3 V. Thus, separation of bending curvature manipulation and AFM scan enables the accurate topographical imaging in Figure. 1e. Since the fabricated free-standing 2D ferroelectric heterostructure has approximately 214.04 nm thickness, the naturally-induced mechanical bending in nanogap was hindered unlike the monolayer and bilayer scale (Supplementary Fig. 1d). As the external shear force should be driven mechanical bending at the nanogap region [Wu, Mei, et al. *Nature Communications* **13.1**

(2022)], the minimum applied force for mechanical bending is measured as 478.32 nN in our work. We thank the reviewer for pointing this out and have revised the manuscript as follows:

[Revised Text] Main text (Page 5):

As shown in Fig. 1e, the nanoscale strain gradient in the free-standing state was precisely controlled with vertical shear strain. The nanoscale bending curvature was independently manipulated at the heterogeneous free-standing gap region through the tip-induced vertical shear strain. The nanoscale bending curvature radius was experimentally obtained as 97.39 nm (480 nN), 38.72 nm (600 nN), and 25.14 nm (720 nN). Following the tip-induced strain engineering, 3D topography was measured with conventional AFM imaging. Thus, separation of tip-induced strain engineering and topography scanning enables the accurate topographical imaging in Fig. 1e. While the thickness of free-standing 2D ferroelectric heterostructure has been measured as 214.04 nm, the naturally-induced mechanical bending at the nanogap was hindered unlike monolayer and bilayer scale, which requires the external force (478.32 nN) to achieve nanoscale bending curvature at the nanogap region.

[Revised Text] Methods (Page 18):

AFM (NX-10, Park Systems, Republic of Korea) measurements were conducted using an ElectriMulti75-G cantilever, with a silver paste electrode selectively deposited at the edge of the sample to induce vertical charge transfer. The ElectriMulti75-G cantilever was calibrated with a tip radius (25 nm), length (225 μm), height (17 μm), width (28 μm), and spring constant (3.3 N/m), which resulted in 320.8 kHz contact resonance frequency. During the tip-induced strain engineering, the inhomogeneous shear strain was vertically controlled with a gradual force limitation (480, 600, and 720 nN), hold time (50 s), and speed (0.5 $\mu\text{m/s}$). After the tip-induced strain engineering, AFM topography was measured with ElectriMulti75-G cantilever, contact mode, scan rate 0.7 Hz, and set point 0.3 V.

[Revised Text] Supplementary Information (Page 2):

0 nN measurement has been additionally added in Supplementary Fig. 1e, f.

Supplementary Fig. 1. Structural characterization of axial nanogap structure. (a) Optical image of the axial nanogap structure, consisting of 500 nm width and 150 nm depth. (b) SEM image of fabricated axial nanogap array. (c) Optical image of free-standing 2D α -In₂Se₃/CuInP₂S₆ heterojunction on the axial nanogap structure and (d) its corresponding cross-sectional TEM image of free-standing 2D α -In₂Se₃/CuInP₂S₆ heterojunction. (e) Line profile and (f) its corresponding CAFM IV curve at the free-standing 2D ferroelectric heterojunction with various applied force, enabling the comparison with free-standing states (0 nN).

Review #1_Q6) In Fig. 1f, the colors of 480nN and 600nN are indistinguishable. I would kindly suggest using a higher-contrast color map.

Response:

Thank you very much for the reviewer’s considerate comment. We recognized that the color distinguishability of 480 nN and 600 nN is too low in Fig. 1f. We have increased the visibility in the revised manuscript as follows. In the revised manuscript, the colormap was replaced to increase distinguishability and visibility.

[Revised Text] Fig. 1f (Page 26):

Fig. 1f has been edited as follows.

Review #1_Q7) How is the nano-wrinkle structure fabricated in Fig. 2? Is it just finding some random wrinkle on the sample?

Response:

We appreciate the reviewer for pointing out the insufficient explanation about the nano-wrinkle structure. We have measured the data included in Fig. 2 at the arbitrarily generated nano-wrinkle spot during the dry transfer process. We thank the reviewer for pointing this out and have revised the sentence as follows:

[Revised Text] Methods (Page 18):

Following α - In_2Se_3 dry transfer, CuInP_2S_6 was arbitrarily transferred above the α - In_2Se_3 position by applying the 70 °C transfer condition, which determined the local freestanding states. Alternatively, the nanowrinkle structure was naturally generated during the dry transfer process (Supplementary Fig. 18).

[Revised Text] Supplementary Information (Page 19):

Supplementary Fig. 18 has been edited as follows.

Supplementary Fig. 18. 3D topography of nanowrinkle structure in transferred 2D ferroelectric materials. Spatial topography image of transferred (a) nanowrinkle CuInP_2S_6 , (b) nanowrinkle In_2Se_3 .

Review #1_Q8) Some confusing statements need to be refined. For example:

[1] “Nonetheless, previous research was completely limited due to the geometrical substrate suspension and anisotropic strain gradient, insufficient strain gradient, since the axial nanogap platform (Fig. 1d) derives the local free-standing state and an efficient strain gradient, unlike the conventional geometrical substrate”.

[2] “To further clarify the localized electron drift, the heterogeneous electron distribution included the suspended area, compressive strain area, and tensile strain area.”

Response:

We appreciate the reviewer for pointing out the insufficient clarification. We thank the reviewer for pointing this out and have revised the sentence as follows:

[Revised Text] Main text (Page 5):

[1] Nonetheless, the previous research of flexoelectric engineering was completely limited due to the geometrical substrate suspension [Das, Saikat, et al. *Nature communications* 10.1 (2019)] and anisotropic strain gradient [Lun, Yingzhuo, et al. *Advanced Materials* 35.36 (2023)], while the axial nanogap platform (Fig. 1d) derives the local free-standing state and an efficient strain gradient, unlike the conventional structural platform.

[Revised Text] Main text (Page 7):

[2] To further clarify the heterogeneous electron distribution, the EFM amplitude at the edge and center area were spatially mapped as -21.70 mV (CuInP₂S₆ nanogap), 13.76 mV (α -In₂Se₃ nanogap), 6.76 mV (CuInP₂S₆ nanowrinkle), and -2.75 mV (α -In₂Se₃ nanowrinkle), which indicates the nano-confined electron distribution within inhomogeneous strain, attributed to the down polarization.

Review #1_Q9) The authors state that “According to the geometric extendable axial nanogap array, wafer-scale spatial confinement of the conductive filaments was achieved,

as shown in Fig. 4b” However, only a 30 um profile was shown in Fig. 4b, not wafer-scale. And what is the force for Fig. 4b?

Response:

We appreciate the reviewer for pointing out the insufficient clarification about the axial nanogap array. To clarify the wafer-scale extendibility of the axial nanogap platform, we have added this additional supplementary figure (Supplementary Fig. 15), which consists of the CAFM line profile of free-standing 2D ferro-ionic memristor on the axial nanogap array and photography image of 6-inch wafer-scale axial nanogap array. Also, 600 nN force was applied in Fig. 4b. We thank the reviewer for pointing this out and have revised the sentence as follows:

[Revised Text] Main text (Page 11):

As shown in Fig. 4b, programmable flexoelectric engineering at the axial nanogap array was realized with an applied force of 600 nN, confirming the robust reproducibility and reliability of tip-induced strain engineering. Regarding the geometric extensibility of the axial nanogap array, the spatial confinement of the conductive filaments can be homogeneously extended to a 6-inch wafer-scale axial nanogap structure (Supplementary Fig. 15).

[Revised Text] Supplementary Information (Page 16):

Supplementary Fig. 15 has been edited as follows.

Supplementary Fig. 15. Geometric extendibility of wafer-scale axial nanogap array.

(a) Photography of 6-inch wafer-scale axial nanogap array. (b) 3D topography and (c) its corresponding CAFM current mapping at the axial nanogap array.

Review #1_Q10) The scale bar is missing in Supplementary Fig. 1a, almost invisible in Supplementary Fig. 3 b, c, e, f

Response:

We apologize for the exclusion of the scale bar plot (Supplementary Fig. 1a) and the invisibility of the scale bar plot (Supplementary Fig. 3 b, c, e, f). To deliver clear information to readers, we have added and substituted the scale bar plot to increase visibility (Supplementary Fig. 1a and Fig. 3 b, c, e, f)

[Revised Text] Supplementary Information (Page 2):

Scale bar plot has been edited in Supplementary Fig. 1.

Supplementary Fig. 1. Structural characterization of axial nanogap structure. (a) Optical image of the axial nanogap structure, consisting of 500 nm width and 150 nm depth. (b) SEM image of fabricated axial nanogap array. (c) Optical image of free-standing 2D α - $\text{In}_2\text{Se}_3/\text{CuInP}_2\text{S}_6$ heterojunction on the axial nanogap structure and (d) its corresponding cross-sectional TEM image of free-standing 2D α - $\text{In}_2\text{Se}_3/\text{CuInP}_2\text{S}_6$ heterojunction. (e) Line profile and (f) its corresponding CAFM IV curve at the free-standing 2D ferroelectric heterojunction with various applied force, enabling the comparison with free-standing states (0 nN).

[Revised Text] Supplementary Information (Page 4):

Scale bar plot has been edited in Supplementary Fig. 3.

Review #1_Q11) There is no definition of CuInP₂S₆ as CIPS. But many figures labeled CIPS.

Response:

We apologize for the usage of the unverified abbreviation “CIPS”. Therefore, we have substituted the unverified abbreviation “CIPS” with CuInP₂S₆ in Supplementary Fig. 4, 5, 6, and 9.

[Revised Text] Supplementary Information (Page 5):

unverified abbreviation “CIPS” has been edited in Supplementary Fig. 4.

[Revised Text] Supplementary Information (Page 6):

unverified abbreviation “CIPS” has been edited in Supplementary Fig. 5.

[Revised Text] Supplementary Information (Page 7):

unverified abbreviation “CIPS” has been edited in Supplementary Fig. 6.

[Revised Text] Supplementary Information (Page 10):

unverified abbreviation “CIPS” has been edited in Supplementary Fig. 9.

Supplementary Fig. 10. Reversible paraelectric threshold voltage shifts within thickness-dependent layer dominance. (a) PFM amplitude and (d) PFM phase of 2D

ferroelectric heterostructure, consisted of α -In₂Se₃ 120.70 nm / CuInP₂S₆ 115.65 nm. (b) PFM amplitude and (e) PFM phase of 2D ferroelectric heterostructure, consisted of α -In₂Se₃ 42.94 nm and CuInP₂S₆ 151.2 nm. (c) PFM amplitude and (f) PFM phase of 2D ferroelectric heterostructure, consisted of α -In₂Se₃ 33.36 nm and CuInP₂S₆ 180.37 nm.

Review #1_Q12) The authors mentioned that CuInP₂S₆ was aligned and transferred to In₂Se₃. Are two straight edges from the two flakes aligned? Or just stack CIPS on top of In₂Se₃ with arbitrary orientation?

Response:

We apologize for causing a confusion to readers. Mechanically-exfoliated CuInP₂S₆ was located and transferred on the α -In₂Se₃ with arbitrary orientation. We have added additional explanation to deliver clear information to readers.

[Revised Text] Methods (Page 18):

Following α -In₂Se₃ dry transfer, CuInP₂S₆ was arbitrarily transferred above the α -In₂Se₃ position by applying the 70 °C transfer condition, which determined the local freestanding states. Alternatively, the nanowrinkle structure was naturally generated during the dry transfer process (Supplementary Fig. 18).

Reviewer (#2)'s COMMENTS:

The authors report a proof-of-concept memristor based on CuInP₂S₆/In₂Se₃ which is programmed using a PFM tip. I found the article to be poorly written and rather difficult to follow. More importantly, I believe the authors give a physical explanation of the mechanisms at play that is not correct, since the flexoelectric coupling seems to have been misunderstood, and the piezoelectric effect (which may be even larger than the flexoelectric effect) is completely overlooked. Also, the double phase transition undergone by the CuInP₂S₆ (CIPS) layer which is partly responsible for the programmability is not

sufficiently backed up by the measurements (or the model) presented by the authors. My recommendation is therefore to not publish the manuscript in Nature Communications.

Review #2_Q1) Flexoelectricity seems to have been misunderstood by the authors, hence the physical mechanism used to explain their observations seems wrong. A compressive or a tensile strain on their own do not induce a change in polarization via a flexoelectric coupling, but rather via a piezoelectric coupling. It is only a strain gradient that induces a change in the polarization of a flexoelectric origin. Instead, the authors often refer to the flexoelectric origin of a polarization in regions of compressive or tensile strain, which therefore leads to wrong statements. The regions under homogeneous strain, like for instance those indicated in Fig. 2a as “downward flexoelectricity” and in 2b as “upward flexoelectricity”, which are around the inflection point of the indented membrane (and hence with a very small or null average strain gradient), should show no net flexoelectricity. (The terms upward and downward flexoelectricity can be misleading and are not well defined). Also, the terms tensile and compressive strain are used for the central region around the PFM tip in the nano-wrinkle and nano-gap geometries, respectively. However, the region around the gap is in both cases under in-plane tensile strain, since the structure is most likely stretched around the central point. Therefore, the whole discussion is very misleading, and some conclusions made on the basis of the sign of the strain in the central region are necessarily wrong, or at least wrongly stated.

Response:

We apologize for the unclarified description of the flexoelectric mechanism. We acknowledge that the use the terminology “*upward / downward*” and “*compressive / tensile*” can cause the misunderstanding to readers. According to the previous CuInP_2S_6 flexoelectricity research [Ming, Wenjie, et al. *Science advances* **8.33** (2022)], the flexoelectricity can be facilitated with a local inhomogeneous strain gradient, which can be explained by the strong correlation of polarization and lattice bending of CuInP_2S_6 . When

there is an upward bending in CuInP_2S_6 , the upper part of the flake is elongated, while the lower part is shortened, and this strain gradient propels Cu^+ ions into the position of upward polarization. When it is bent downward, the opposite occurs, and this is precisely what we know as the flexoelectric effect [Neumayer, Sabine M., et al. *ACS nano* **16.2** (2022)]. As the terminology “*compressive strain / tensile strain*” has been selected to express the topographical elongation (tensile strain) and topographical shorten (compressive strain), the terminology “*tensile strain*” is replaced with the “**upward bending**” and the terminology “*compressive strain*” is replaced with the “**downward bending**” to generally clarify the flexoelectricity direction with mechanical bending direction.

[Revised Text] Main text

All terminology “*tensile strain*” is replaced with the “**upward bending**” and the terminology “*compressive strain*” with the “**downward bending**”.

[Revised Text] Fig. 2 (Page 27):

Fig. 2 has been edited as follows.

Review #2_Q2) Equally importantly: CIPS is piezoelectric, and hence the piezoelectric response should play an important role in the picture – it is in fact the lowest order non-zero term in the polarization-strain coupling for this material and should therefore be considered. Still, the authors have essentially ignored this possibility. I believe the authors should consider piezoelectric effects when explaining their findings or explain why they can be ruled out.

Response:

We apologize for the insufficient explanation on the piezoelectric effects. As the piezoelectric materials can be electrically polarized within uniform strain, and conversely experience a strain in response to an applied electric field which is in proportion to the strength of the field. Thus, in contrast to piezoelectricity, flexoelectricity refers specifically

to polarization due to local strain that changes from point to point in the material, whereby it exhibits a spontaneous electrical polarization induced by inhomogeneous strain gradient (splay- or bent-deformations). This different origin typically indicates the flexoelectric effect insignificant relative to piezoelectricity at the microscales, due to mechanical restrictions on forming large strain gradients. After the tip-induced strain engineering, vertical compressive strain (piezoelectricity) and inhomogeneous strain gradient (flexoelectricity) can be concurrently induced at the free-standing region. Nonetheless, the flexoelectricity is an inherently size-dependent phenomenon. When the flake thickness decreases to the nanoscale dimension, the inhomogeneous strain gradient dominantly overwhelms the vertical compressive strain, which are both induced by tip-induced vertical shear strain [Park, Sung Min, et al. *Nature nanotechnology* **13.5** (2018)] [Das, Saikat, et al. *Nature communications* **8.1** (2017)], [Molina-Luna, Leopoldo, et al. *Nature communications* **9.1** (2018)]. Therefore, the CuInP_2S_6 piezoelectric effect can be excluded from this discussion because the flexoelectricity dominance overwhelms the piezoelectricity at nanoscale systems. We thank the reviewer for pointing this out and have revised the original text with the additional explanation as follows:

[Revised Text] Main text (Page 7):

Furthermore, the heterogeneous origin of piezoelectricity and flexoelectricity has been discussed. Although the piezoelectric materials can be electrically polarized within uniform strain, flexoelectricity refers specifically to polarization due to local strain that changes according to the strain gradient. After the tip-induced strain engineering, vertical compressive strain (piezoelectricity) and inhomogeneous strain gradient (flexoelectricity) can be concurrently induced at the free-standing region. Nonetheless, the flexoelectricity is an inherently size-dependent phenomenon. When the flake thickness decreases to the nanoscale dimension, the inhomogeneous strain gradient dominantly overwhelms the vertical compressive strain, which are both induced by tip-induced vertical shear strain (Supplementary Fig. 7). Therefore, the piezoelectric effect of the free-standing 2D

heterostructure can be excluded from this discussion, as it can be attributed from the nanoscale flexoelectric dominance [Park, Sung Min, et al. *Nature nanotechnology* 13.5 (2018)] [Das, Saikat, et al. *Nature communications* 8.1 (2017)], [Molina-Luna, Leopoldo, et al. *Nature communications* 9.1 (2018)].

[Revised Text] Supplementary Information (Page 8):

Supplementary Fig. 7 has been edited as follows.

Supplementary Fig. 7. Schematic illustration of piezoelectricity and flexoelectricity of free-standing 2D ferroelectric heterostructure. Schematic of (a) tip-induced strain engineering. Spatial strain distribution of (b) piezoelectricity and (c) flexoelectricity, which is induced with homogeneous vertical compression (piezoelectricity) and inhomogeneous strain gradient (flexoelectricity).

Review #2_Q3) The phase transitions shown in Fig. 5 (from ferroelectric to ferroionic to paraelectric) seem to be deduced only from the I-V curve shown in Fig. 5a. However, other mechanisms could be at play here (rupture of the sample due to indentation, deposition of atoms from the tip onto the sample, ...). I believe that the authors should back this statement with further measurements – for example, if the ferroelectric phase is truly ferroelectric, they could measure ferroelectric hysteresis loops (P-E) with the PFM. And when it becomes paraelectric, it should show a characteristic paraelectric behavior on a P-E loop.

Response:

We apologize for the unclear description on the ferroelectric hysteresis analysis. To further clarify the ferroelectric hysteresis, the hysteresis (P-E) loop was spatially mapped with nanogap structure and its corresponding PFM spectroscopy. Regardless of voltage range, the suspended region homogeneously exhibits the ferroelectric phase, while the nanogap region indicates the heterogeneous phase between -5 V–5 V sweep range (ferro-ionic phase) and -10 V–10 V sweep range (paraelectric phase). We have added the spatial ferroelectric hysteresis mapping with nanogap topography in the supplementary (Supplementary Fig. 12) Information to clarify the CuInP_2S_6 phase transition with its corresponding I-V curve.

[Revised Text] Main text (Page 10):

Additionally, the ferroelectric hysteresis loop was spatially mapped according to the axial nanogap topography to clarify the CuInP_2S_6 phase transition. Regardless of voltage range, the suspended region exhibited a homogeneous ferroelectric phase, whereas the nanogap region was observed to have a heterogeneous phase, exhibiting ferro-ionic and paraelectric phases within the respective ranges of -5–5 V and -10 V–10 V (Supplementary Fig. 12).

[Revised Text] Supplementary Information (Page 13):

Supplementary Fig. 12 has been edited as follows.

Supplementary Fig. 12. Spatial PFM hysteresis mapping of axial nanogap structure.

(a) 3D topography of axial nanogap structure and PFM hysteresis mapping within (b) -5 V–5 V sweep range and (c) -10 V–10 V sweep range.

Review #2_Q4) The geometry of the whole device is not clear to me from Fig.1. In panel 1c, where does the tip make the indentation? It seems like there is no place where the CIPS layer is accessible.

Response:

We apologize for the lack of description on the free-standing 2D ferro-ionic memristor schematic. Since the memristor has been organized within large-scale cross-bar arrays to implement the efficient in-memory computing architecture, the memristor cross-bar array is illustrated in Fig. 1c (top). To manipulate the mechanical bending at the memristor cross-bar array, a conductive AFM tip (Pt) was implemented for site-selective strain engineering. Therefore, the free-standing 2D ferro-ionic memristor device structure was illustrated in

Fig. 1c (bottom). We have replaced Fig. 1c with a detailed device structure to deliver accurate information to readers.

[Revised Text] Main text (Page 5):

Additionally, device construction of 2D ferro-ionic memristor was exhibited with 3D schematic illustration for structural clarification and 3D stacking order (Fig. 1c). As illustrated in Fig. 1c (top), the memristor has been organized at the crossbar arrays to enable the mechanical bending within conductive AFM tip (Pt), which can act as a top electrode and strain source, was vertically manipulated as illustrated in Fig. 1c (bottom).

[Revised Text] Fig. 1c (Page 26):

Fig. 1c has been edited as follows.

Review #2's minor concerns:

[A] Shear strain and bending curvature values are given in units of nN, but those are units of force and hence cannot be the right units for either of those quantities.

Response:

We apologize for the usage of nN unit in bending curvature. To deliver clear information to readers, we have replaced the “*bending curvature*” with the “**bending curvature radius**” as follows.

[Revised Text] Main text (Page 6):

The nanoscale bending curvature radius was experimentally obtained as 97.39 nm (480 nN), 38.72 nm (600 nN), and 25.14 nm (720 nN).

[B] I could not understand several sentences in the manuscript, where the authors seem to be misunderstanding concepts. These are the most obscure to me:

- “The local flexoelectric energy storage results in 5.76×10^2 -fold increased maximum current was observed within vertical shear strain 720 nN”

Response:

Thank you very much for pointing out the missing explanations in the main text. We thank the reviewer for pointing this out and have revised the sentence as follows:

[Revised Text] Main text (Page 10):

Since the ferro-ionic conduction was spatially generated by nanoscale mechanical bending, robust ferro-ionic current amplification can be site-selectively observed. While the current has been measured as 4.13 nA without strain, the current level is significantly increased with tip-induced shear strain as 93.26 nA (480 nN), 497.67 nA (600 nN), and 2380.84 nA (720 nN). Within 720 nN of force, a 5.76×10^2 -fold maximum current

enhancement was clearly observed, corresponding to the ratio of 2380.84 nA (720 nN) to 4.13 nA (W/O strain).

- “the flexoelectric energy conversion induced an internal electrical field”

Response:

Thank you very much for pointing out the insufficient explanations in main text. We thank the reviewer for pointing this out and have revised the sentence as follows:

[Revised Text] Main text (Page 7):

When the inhomogeneous strain induced the nanoscale lattice bending, which locally activates the internal flexoelectric field in the material, we observed local polarization switching and carrier redistribution.

- “the heterogeneous electron distribution included the suspended area”

Response:

Thank you very much for pointing out the insufficient explanations in main text. We thank the reviewer for pointing this out and have revised the sentence as follows:

[Revised Text] Main text (Page 7):

To further clarify the heterogeneous electron distribution, the EFM amplitude at the edge and center area were spatially mapped as -21.70 mV (CuInP₂S₆ nanogap), 13.76 mV (α -In₂Se₃ nanogap), 6.76 mV (CuInP₂S₆ nanowrinkle), and -2.75 mV (α -In₂Se₃ nanowrinkle), which indicates the nano-confined electron distribution within inhomogeneous strain, attributed to the down polarization.

- “flexoelectric energy storage in free-standing α -In₂Se₃/CuInP₂S₆ heterostructure leads to 1.3272 V ferro-ionic conduction shift”

Response:

Thank you very much for pointing out the insufficient explanations in main text. We thank the reviewer for pointing this out and have revised the sentence as follows:

[Revised Text] Main text (Page 9):

In contrast, the flexoelectric field in the free-standing α -In₂Se₃/CuInP₂S₆ heterostructure reduces the ferro-ionic conduction V_{th} from 5.36 V to 3.24 V, regarding the polymorphicity of CuInP₂S₆ in the consistent voltage range.

- “As flexoelectricity indicates the inherent scale dependence, flexoelectricity dominance emerges at nanoscale material dimensions rather than microscale material dimensions”

Response:

We apologize for the insufficient explanation on the scale effects of flexoelectricity. As the scale effects in flexoelectricity has been considered in “Review #2 Q2”, we thank again the reviewer for pointing this out and have revised the original text with the additional explanation as follows:

[Revised Text] Main text (Page 7):

Nonetheless, the flexoelectricity is an inherently size-dependent phenomenon, which dominantly overwhelms the piezoelectric response at nanoscale dimension. Therefore, the piezoelectric effect of the free-standing 2D heterostructure can be excluded from this discussion, as it can be attributed to the nanoscale flexoelectric dominance [Park, Sung Min, et al. *Nature nanotechnology* 13.5 (2018)] [Das, Saikat, et al. *Nature*

communications **8.1** (2017)], [Molina-Luna, Leopoldo, et al. *Nature communications* **9.1** (2018)].

Some other sentences require further explanations or context, which is not provided:

- “According to previous ferro-ionic research, the spatial modulation of conductive filaments was completely blocked”. This statement asks for further elaboration, plus a reference to the previous research mentioned.

Response:

We apologize for the missing references. To deliver clear information to readers, we have added the previous research about stochastic conductive filaments [Lun, Yingzhuo, et al. "Ultralow Tip-Force Driven Sizable-Area Domain Manipulation through Transverse Flexoelectricity." **35.36** (2023): 2302320.], [Jiang, Xingan, et al. *Advanced Functional Materials* **33.40** (2023)], [Ming, Wenjie, et al. *Science advances* **8.33** (2022)], [Liu, Yongtao, et al. *ACS nano* **17.21** (2023)]. These previous works only control its ferroelectric domain switching with its flexoelectricity. While the strain gradient, which was induced for flexoelectricity, were totally not enough for the activation of CuInP_2S_6 ferro-ionic conduction, the site-selective manipulation of conductive filaments has been hindered.

[Revised Text] Main text (Page 11):

According to previous ferro-ionic studies, the flexoelectric engineering of CuInP_2S_6 has been actively reported [Lun, Yingzhuo, et al. "Ultralow Tip-Force Driven Sizable-Area Domain Manipulation through Transverse Flexoelectricity." **35.36** (2023): 2302320.], [Jiang, Xingan, et al. *Advanced Functional Materials* **33.40** (2023)], [Ming, Wenjie, et al. *Science advances* **8.33** (2022)], [Liu, Yongtao, et al. *ACS nano* **17.21** (2023)]. These previous works are only available to control its ferroelectric domain switching with mechanical switching methods. While the strain gradient, which was reported in previous

works, were completely insufficient to activate the CuInP_2S_6 ferro-ionic conduction, the spatial modulation of conductive filaments has been completely blocked.

• “According to equation (1), it is evident that both the microscale and nanoscale strain gradients exhibit flexoelectric polarization amplification within compressive strain, while the localized polarization distribution and field intensity of the microscale strain gradient is significantly lower than the nanoscale strain gradient”. No local considerations can be derived -or at least they are not evident at all- from Eq.1, which gives a macroscopic Gibbs free energy. Also, compressive strain itself does not induce a flexoelectric polarization (rather a change in strain does). Finally no distinction between microscale and nanoscale behavior can be deduced from Eq. (1).

Response:

We apologize for the insufficient explanation and unclarified contextualization of the heterogeneity of micro / nanoscale strain gradient. To deliver clear information to readers, we have added the previous nanogap research about micro / nanoscale strain gradient [Lee, Hyeongwoo, et al. *Science advances* 8.5 (2022)] and coupling with the flexoelectric field.

[Revised Text] Main text (Page 14):

According to Equation (1), the strain gradients exhibited amplified polarization during downward bending. Moreover, the local polarization redistribution and amplification were theoretically supported by performing Gibbs free energy modeling. The simulation revealed that, as the bending curvature increases, the amplitude of the flexoelectric field proportionally increases, which can be further clarified by the previous nanogap research including a microscale strain gradient that also yielded a significantly lower bending curvature than the nanoscale strain gradient [Lee, Hyeongwoo, et al. *Science advances* 8.5 (2022)].

[C] The figures for the (thicknesses?) of the layers conforming the system are given in nm with two and sometimes three significant decimal numbers, implying sub-angstrom resolution, but I believe the methods used do not have that actual resolution. Many other figures in the manuscript also imply very high resolutions (voltages, angles), but I am no expert in typical PFM output signals and cannot judge the validity of those implied precisions.

Response:

We appreciate the reviewer's valuable comment to improve the completeness of our manuscript. AFM measures the various parameters (Z height, work function, electrical potential, current, hardness, modulus, piezoelectric amplitude, piezoelectric phase) with cantilever dynamics, which can detect the surface characteristics. According to the previous AFM research [Krieg, Michael, et al. *Nature Reviews Physics* 1.1 (2019)], the laser beam at the cantilever is reflected to the position sensitive photodetector (PSPD). During the AFM scanning, PSPD detects the vertical / lateral movement of the laser beam, and the movement of the laser beam is saved as pixel data and converted to the 2D pixel array. Hence, the implied decimal place is attributed to the measured cantilever dynamics and its corresponding laser beam motions, exactly extracted from the tip-sample interaction. To exclude the measurement uncertainty, we have rounded each experimental value down to 2 decimal places.

[Revised Text] Main text (Page 14):

[D] Fig. 5f is missing the legend for the color bar. Also, I believe it would be more illustrative to show a map of the strain gradient in the sample, rather than the actual strain. This would show where flexoelectricity should induce larger changes in the polarization.

Response:

We apologize for the missing color legend plot in Fig. 5f. To deliver clear information to readers, we have added color legend to increase visibility (Fig. 5f). Nonetheless, we think that there was a misunderstanding by the reviewer about strain gradient mapping. We already have plotted the strain gradient mapping and its corresponding cross-sectional view in Fig. 5f, not an actual strain distribution.

[Revised Text] Fig. 5f (Page 30):

Fig. 5f. has been edited as follows.

[E] I could not understand the following sentence: “the force-distance curve and current-distance curve indicates the coupling of the vertical shear strain and strain gradient in the free-standing area, using programmable flexoelectric energy conversion”. The indentation with the tip induces vertical (shear?) strain locally, which results in a strain gradient between the region around the tip and the regions away from it. I do not understand what coupling the authors are thinking of and what role does the programmable flexoelectric energy conversion play here.

Response:

We apologize for the insufficient description of the terminology “programmable” and “flexoelectric energy conversion”, which may cause confusion for readers. In this work, the terminology “programmable” was used to describe the mechanically controllable ferro-ionic conduction. As similar research, the tip-induced vertical shear strain has been recently reported to produce the programmable graphene nano bubbles [Jia, Pengfei, et al. *Nature Communications* **10**.1 (2019)] and the programmable quantum emission [Koo, Yeonjeong, et al. *Advanced Materials* **33**.17 (2021)], focusing that it was mechanically controlled by vertical AFM tip engineering. In our case, the terminologies of “programmable” can be used to imply the functionalized correlation of current with respect to the scale of strain gradient. Moreover, flexoelectric energy conversion systems have been reported [Yoon, Chongsei, et al. *ACS Energy Letters* **8**.11 (2023)], [Bhaskar, Umesh Kumar, et al. *Nature nanotechnology* **11**.3 (2016)], converting the inhomogeneous mechanical strain to the flexoelectric field. Hence, we believe the terminologies of “programmable” and “flexoelectric energy conversion” are quite reasonable to indicate this generalizable approach. To deliver clear information to readers, we have added additional explanation to clarify the terminology “programmable” and “flexoelectric energy conversion”.

[Revised Text] Main text (Page 9):

Additionally, the force-distance curve and topography line profile indicate the accurate control of 3D tip positioning techniques in three-dimension scale (Supplementary Fig. 9. a, b). Regarding the 3D tip positioning techniques, inhomogeneous mechanical strain can generate the programmable flexoelectric field, which implies the functionalized correlation of ferro-ionic current with respect to the scale of strain gradient.

[F] The abstract is hard to understand because it lacks contextualization. For example, the “external conditions” the authors refer are only later explained in the text, but in the abstract they seem something special and unexplained.

Response:

We apologize for the insufficient contextualization of the abstract. We have added an additional explanation of “external conditions” to deliver clear information to readers.

[Revised Text] Abstract (Page 2):

As CuInP_2S_6 exhibits a ferroelectric phase with insulating properties at room temperature, the external temperature and electrical field should be required to activate the ferro-ionic conduction.

[Revised Text] Abstract (Page 2):

Herein, free-standing 2D ferroelectric heterostructure was mechanically manipulated for nano-confined conductive filaments growth in local free-standing 2D ferro-ionic memristor.

[Revised Text] Abstract (Page 2):

According to the localized flexoelectric energy storage,

[G] I do not understand what are the “heterogeneous” and “homogeneous” field directions (page 7, towards the end).

Response:

We apologize for the unclear description of the terminology “*heterogeneous* / *homogeneous*”, which may cause confusion for readers. This refers to the case where the flexoelectricity direction is opposite to the direction of ferro-ionic conduction. The term

“*heterogeneous*” has been replaced with “attenuation direction” and “*homogeneous*” with “compensation direction” to clarify the flexoelectricity direction with Cu⁺ ion dynamics.

[Revised Text] Main text (Page 8):

The downward (upward) lattice bending exhibits the compensation (attenuation) direction of Cu⁺ ion dynamics, which implies that the ferro-ionic conduction threshold voltage (V_{th}) can be reversibly controlled according to the lattice bending direction.

Reviewer (#3)'s COMMENTS:

This article reported a programmable flexoelectric energy storage platform in free-standing 2D ferro-ionic memristor. This universal nanogap platform can provide the extendable geometric solution for nanoscale flexoelectric energy conversion efficiency and reliable neuromorphic device. Although CuInP2S6 has been reported many times for the neuromorphic memristor, this manuscript offered a new memristor via a novel approach, the results are interesting, the measurement data are solid. I recommend publication after minor revision.

Response:

We thank the reviewer for acknowledging the novelty and significance of our work. According to the raised suggestion, we have performed additional experiments with corresponding revisions to our revised manuscript.

Review #3_Q1) Temperature-independence ferro-ionic conduction behavior is very important for the device application. Therefore, it would be better if authors can offer some temperature-independent conduction or current behavior.

Response:

We appreciate the reviewer's valuable comment to improve the completeness of our manuscript. To further clarify the temperature independence, additional experiments, references, and corresponding revisions were inserted into our revised manuscript. Recently, Jiachao Zhou et al. [*Advanced Materials* **35.38** (2023)] classified the four heterogeneous phases in CuInP_2S_6 , namely frozen-in polarization state, ferroelectric polarization state, Cu ions hopping state, and conductive filament state. According to the reported CuInP_2S_6 phase transition diagram with temperature and an external electric field, we have split the bias sweep range as -10 V–10 V and -20 V–20 V, while the temperature condition has been distributed as 315 K, 330 K, 350 K, and 400 K for measuring the temperature effects in our fabricated nanogap device. To measure the temperature effects, the free-standing 2D nanogap device was additionally fabricated (Supplementary Fig. 14a and b), which is constructed with electrode gap (1.66 μm) and electrode width (4.18 μm). In detail, the metal electrodes are patterned with electron-beam lithography (JSM-7001F SEM system, JEOL) followed by Ti/Au (20/50 nm) metal deposition via electron-beam evaporation. Ferro-ionic conduction behavior results in a homogeneous ferro-ionic phase with temperature-independency (Supplementary Fig. 14c and d). However, topographical Cu^+ ion extraction reproducibly activates in 400 K condition, which is attributed to the paraelectric phase transition of CuInP_2S_6 . Since our temperature-independent conduction behavior accurately correlates with previous CuInP_2S_6 research [*Advanced Materials* **35.38** (2023)], temperature-independence and reproducible ferro-ionic conduction characteristics are experimentally verified under the 400 K condition.

[Revised Text] Main text (Page 10):

While the temperature-independency of ferro-ionic conduction is also significant for device applications, the temperature effects of ferro-ionic conduction have been further verified under 315 K, 330K, 350 K, and 400 K conditions. According to the previous research [*Advanced Materials* **35.38** (2023)], the CuInP_2S_6 phase has been classified as four heterogeneous states, i.e., the frozen-in polarization state, ferroelectric polarization state,

Cu⁺ ion hopping state, and conductive filament state. Because a temperature-dependent CuInP₂S₆ phase transition has been reported at 10² MV/m and 300 K [Advanced Materials 35.38 (2023)], the free-standing 2D nanogap device also exhibited temperature-independent homogeneous ferro-ionic conductive behavior (Supplementary Fig. 14) under the 400 K condition. Moreover, the IV hysteresis behavior in -10 V–10 V sweep range is inhomogeneous, which is attributed to the ferro-ionic phase transition. While the -10 V–10 V range possess the possibility of ferro-ionic phase transition within temperature variation, the -20 V–20 V sweep range clearly exceed the ferro-ionic conduction V_{th}. Additionally, the topographical Cu⁺ ion extraction reproducibly was activated under the 400 K condition, which is attributed to the paraelectric phase transition of CuInP₂S₆. Consequently, temperature-independence and reproducible ferro-ionic conductive behavior have been experimentally derived under the 400 K condition.

[Revised Text] Supplementary Information (Page 15):

Supplementary Fig. 14 has been edited as follows.

Supplementary Fig. 14. Temperature effects in free-standing 2D ferroelectric nanogap device. (a) OM image and (b) 3D topography of free-standing 2D nanogap

device, which is fabricated for temperature effects. Within various temperature conditions, ferro-ionic conduction mapping at (c) -10 V–10 V sweep range, and (d) -20 V–20 V sweep range, exhibiting the temperature-independent conduction in 315K, 330K, and 350K.

=====

We very appreciate the reviewers' all the valuable comments, which greatly helped us to improve this manuscript.

REVIEWER COMMENTS

Reviewer #1 (Remarks to the Author):

The manuscript has been improved by revision. All my concerns are addressed carefully. The manuscript is now acceptable for publication.

Reviewer #2 (Remarks to the Author):

The authors have kindly addressed all of my minor concerns, and also my last two of major concerns (about the phase transitions and the device geometry). However, I am afraid that the first two major concerns of mine have not been successfully addressed yet.

(1) Some of the claims in the article still seem to be based in a misunderstanding of flexoelectricity. In particular, the authors identify a central region of upward bending in the nano-wrinkle surrounded by regions of downward bending, indicating nearby regions with opposite strain gradient, and therefore opposite polarization change induced by the flexoelectric effect. Nevertheless this picture is inconsistent with the simulated strain gradients shown in Figure 5f (which are only clear now that the color scale shows its units, and the authors mention in their response that the heatmap corresponds to the strain gradient and not strain). In the simulation, the strain gradient is always of the same sign across the sample, which would induce a change in polarization via flexoelectricity of the same sign across the nano-wrinkle. This discrepancy should be addressed.

Also (much more anecdotal but pointing in the direction of a misunderstanding of flexoelectricity by the authors), the reference given in their response to explain what the authors “know as flexoelectric effect” does not mention flexoelectricity throughout the text (Neumayer et al, ACS Nano 16, 2).

(2) I agree with the authors that the length scales may play a role when comparing piezoelectric and flexoelectric effects. However, I believe that the arguments for a negligible piezoelectric contribution to the polarization changes are not sufficiently convincing. Without a quantification of both contributions, which is absent in the manuscript, it is not clear which one dominates, and hence the claim that flexoelectricity dominates over piezoelectricity is not justified. The references pointing at similar effects mentioned in the authors' response and in the updated manuscript correspond to studies on materials that are structurally very different and may show a very different flexoelectric response than CuInP2S6 (BiFeO3, SrTiO3, sodium bismuth titanate–0.25 strontium titanate), so I believe they are not sufficient to justify this point. What's more, the piezoelectric coefficient of CuInP2S6 has been reported to be particularly large (Nano Energy 99, 107371), and the (homogeneous) strains induced by the PFM tip on a freestanding membrane should be rather large. Therefore, the piezoelectric contribution is expected to be large too.

Reviewer #3 (Remarks to the Author):

the authors responded point by point in the revised version. as far as my comment part, the authors offered the effect of ferro-ionic conduction on the temperature under 315 K, 330K, 350 K, and 400 K conditions. from the results, it can be seen clearly that temperature impacts the I-V curves in the temperature span from 315 to 400K, therefore, i think, ferro-

ionic conductive behavior is temperature-dependent. however, the authors use the word "temperature-independence", i was confused, was it a mistake or not?

[Revision #2] Response to the *Nature Communications* Reviewers' Comments

Dear reviewers,

We the authors deeply thank the reviewers for their thorough reading of our manuscript and the constructive comments. With regard to the reviewers' concerns raised, we have revised the manuscript and supporting information based on the reviewers' comments. The reviewers' comments appear in black, and the authors' responses in blue. In the revised manuscript, the changes with respect to the previous version are highlighted.

Sincerely,

Taesung Kim

=====

Point-by-point Response

=====

Reviewer (#1)'s COMMENTS:

The manuscript has been improved by revision. All my concerns are addressed carefully. The manuscript is now acceptable for publication.

Response:

We thank the reviewer for the helpful comments to improve our manuscript. We are glad that the reviewer satisfactorily reads our answer and finds our manuscript suitable for publication.

Reviewer (#2)'s COMMENTS:

The authors have kindly addressed all of my minor concerns, and also my last two of major concerns (about the phase transitions and the device geometry). However, I am afraid that the first two major concerns of mine have not been successfully addressed yet.

Review #2_Q1)

Some of the claims in the article still seem to be based in a misunderstanding of flexoelectricity. In particular, the authors identify a central region of upward bending in the nano-wrinkle surrounded by regions of downward bending, indicating nearby regions with opposite strain gradient, and therefore opposite polarization change induced by the flexoelectric effect. Nevertheless, this picture is inconsistent with the simulated strain gradients shown in Figure 5f (which are only clear now that the color scale shows its units, and the authors mention in their response that the heatmap corresponds to the strain gradient and not strain). In the simulation, the strain gradient is always of the same sign across the sample, which would induce a change in polarization via flexoelectricity of the same sign across the nano-wrinkle. This discrepancy should be addressed.

Also (much more anecdotal but pointing in the direction of a misunderstanding of flexoelectricity by the authors), the reference given in their response to explain what the authors “know as flexoelectric effect” does not mention flexoelectricity throughout the text (Neumayer et al, ACS Nano 16, 2).

Response:

We apologize for our mistakes and thank the reviewer for pointing out it. To clearly deliver this information, we have thoroughly checked the simulation part and revised the nanogap geometry with coexistence of piezoelectric field and flexoelectric field. Regarding the specific concern of Fig. 5f, the top view of strain distribution and cross-sectional field distribution in nanogap geometry has been provided in the Fig. 5f. As **Reviewer (#2)** mentioned, heterogeneous flexoelectric field of nanogap and nanowrinkle has been additionally discussed with PFM experiments. One more thing to be clarified is that the formation of mechanical bending in free-standing area needs to be propagated from top to bottom surface of the layer (conventionally expressed as vertical direction or Z axis), not center to surround, to generate the dominant flexoelectricity effect. According to the **Review #2** comments, we have considered both spatial strain distribution and coexistence of piezoelectricity and flexoelectricity, which dominantly contributes to generate the

flexoelectricity in Fig. 5f: [1] suspension force of the substrate at edge spot, [2] bending force by the film elasticity at center of the layer. [1] suspension force of the substrate at edge spot, [2] bending force by the film elasticity at center of the layer. The combination of these two different direction forces induces different direction of flexoelectricity via upward/downward bending at center/surround position. However, our simulation result implements only one bending state with one bending force to focus on the mechanism of flexoelectricity, not overall bending state or generation of ferroelectricity at nano-gap/wrinkle state. Additionally, its corresponding revisions were inserted to further clarify the flexoelectric field of the CuInP_2S_6 nanowrinkle structure (Supplementary Fig. 6). Additionally, inserted reference for describing the flexoelectricity [Neumayer, Sabine M., et al. ACS nano 16.2 (2022)] has been replaced. Moreover, we apologize for the confusing terminology “*strain gradient*”, which may cause confusion for readers. Hence, all terminology “*strain gradient*” has been replaced with the “**mechanical bending**” to generally clarify the flexoelectricity direction with mechanical bending direction. We thank the reviewer for pointing this out and have revised the manuscript as follows.

[Revised Text] Figure 5 (Page 31):

Fig. 5 has been edited as follows.

Fig. 5: Theoretical validation of Cu^+ ion dynamics within nanoscale flexoelectric energy storage. (a) Full I-V curve of a single 2D ferro-ionic memristor and (b) its corresponding energy band diagram of (b) upward $\alpha\text{-In}_2\text{Se}_3$ / downward CuInP_2S_6 , (c) downward $\alpha\text{-In}_2\text{Se}_3$ / downward CuInP_2S_6 , (d) downward $\alpha\text{-In}_2\text{Se}_3$ / upward CuInP_2S_6 , and (e) upward $\alpha\text{-In}_2\text{Se}_3$ / upward CuInP_2S_6 . (f) Strain calculation of the free-standing 2D ferroelectric heterostructures with consideration of piezoelectric field and flexoelectric field, exhibiting the cross-sectional distribution of the (g) domain and (h) strain. Theoretical evaluation of flexoelectric energy storage, resulting in the (i) bias-polarization hysteresis and (j) strain-polarization hysteresis.

[Revised Text] Main text (Page 15):

As shown in Fig. 5f–j, a free-standing 2D ferroelectric heterostructure was modeled using experimental bending curvatures to accurately quantify the flexoelectric energy conversion, which accurately matches with previous flexoelectric simulation results [Yang, Weijie, et al. *The Journal of Physical Chemistry Letters* **14.2** (2023)], [Jiang, Xingan, et al. *Advanced Functional Materials* **33.40** (2023)], [Liu, Yongtao, et al. *ACS nano* **17.21** (2023)], [Ming, Wenjie, et al. *Science advances* **8.33** (2022)].

[Revised Text] Main text (Page 8):

To further clarify the flexoelectric dominance in CuInP_2S_6 with nanoscale mechanical bending, flexoelectric field distribution in nanowrinkle has been observed in Supplementary Fig. 6a, b, and c. As the nanowrinkle structure was generated without tip-induced vertical shear strain, internal polarization of nanowrinkle has been dominantly generated from the local lattice bending and its corresponding flexoelectricity. As shown in Supplementary Fig. 6b and c, nanowrinkle PFM amplitude inversely correlates with calculated surface potential of downward flexoelectricity [Jiang, Xingan, et al. *Advanced Functional Materials* **33.40** (2023)], while the nanowrinkle PFM phase linearly corresponds with surface potential of downward flexoelectricity [Jiang, Xingan, et al. *Advanced Functional Materials* **33.40** (2023)]. Moreover, tip-induced strain engineering has been conducted within substrate suspension to further minimize the flexoelectric effects. Within substrate suspension, applied tip force exhibits linear correlation with the distance (Supplementary Fig. 6d). Therefore, the applied tip force (480 nN, 600 nN, 720 nN) is insufficient to activate the lattice bending in suspended substrate. In this case, plastic deformation can be completely hindered, which effectively minimizes the flexoelectric field effect. The hysteresis variation was also spatially mapped in suspended substrate with each applied tip force, resulting the absence of hysteresis variation of suspended CuInP_2S_6 (Supplementary Fig. 6e). Hence, the flexoelectric dominance was clearly observed in nanoscale geometry, constructed as the nanowrinkle structure and the suspended substrate [Park, Sung Min, et al. *Nature*

nanotechnology 13.5 (2018)] [Das, Saikat, et al. *Nature communications* 8.1 (2017)], [Molina-Luna, Leopoldo, et al. *Nature communications* 9.1 (2018)].

[Revised Text] Supplementary Information (Page 7):

Supplementary Fig. 6 has been added as follows.

Supplementary Fig. 6. Observation of flexoelectric dominance with nanoscale mechanical bending. Schematic of (a) upward lattice bending in nanowrinkle structure. Spatial distribution of (b) PFM responses and (c) its corresponding line profile, which exhibits the flexoelectric dominance in wrinkle apex area. (d) FD curve measurements at the suspended substrate. (e) PFM hysteresis mapping of substrate suspension, which effectively hinders the lattice bending and its corresponding hysteresis variation with each applied force.

[Revised Text] Response letter to Reviewers (Page 20):

When it is bent downward, the opposite occurs, and this is precisely what we know as the flexoelectric effect [Yang, Weijie, et al. *The Journal of Physical Chemistry Letters* **14.2** (2023)], [Jiang, Xingan, et al. " *Advanced Functional Materials* **33.40** (2023)], [Liu, Yongtao, et al. *ACS nano* **17.21** (2023)], [Ming, Wenjie, et al. *Science advances* **8.33** (2022)].

[Revised Text] Main text

All terminology “*strain gradient*” has been replaced with the “**mechanical bending**” to generally clarify the flexoelectricity direction with mechanical bending direction.

[Revised Text] Supplementary Information (Page 9):

All terminology “*strain gradient*” in Supplementary Fig. 8 has been replaced with the “**mechanical bending**” to generally clarify the flexoelectricity direction with mechanical bending direction.

Supplementary Fig. 8. Correlation of nanoscale mechanical bending and flexoelectric ferro-ionic current. (a) current-distance curve and (b) force-distance curve of free-standing 2D α -In₂Se₃/CuInP₂S₆ heterostructure. (c) PFM amplitude exhibits the ferro-ionic phase without nanoscale mechanical bending, while the paraelectric phase occurred within nanoscale mechanical bending.

[Revised Text] Figure 3 (Page 29):

All terminology “*strain gradient*” in Fig. 3 has been replaced with the “mechanical bending” to generally clarify the flexoelectricity direction with mechanical bending direction.

Fig. 3: Self-powered 2D ferro-ionic conduction via spatial flexoelectric nanomanipulation. (a) Ferro-ionic phase transition mechanism in the 2D α - $\text{In}_2\text{Se}_3/\text{CuInP}_2\text{S}_6$ heterostructure, which indicates the flexoelectric energy storage. (b) Cross-sectional TEM image and its corresponding SAED pattern of the bended In_2Se_3 lattice structure, which induces the nanoscale flexoelectricity. (c) PFM amplitude hysteresis behavior with nanoscale mechanical bending, indicating the conduction threshold reduction. Self-powered system in 2D ferro-ionic conduction, which was experimentally verified with (d) semi log I-V curve, (e) I_{max} distribution, and (f) ratio of I_{max} , with strain to I_{max} , without strain. (g)

Spatial I_{\max} mapping in free-standing 2D α - $\text{In}_2\text{Se}_3/\text{CuInP}_2\text{S}_6$ heterostructure, which enables the sub-50nm flexoelectric manipulation.

Review #2_Q2)

I agree with the authors that the length scales may play a role when comparing piezoelectric and flexoelectric effects. However, I believe that the arguments for a negligible piezoelectric contribution to the polarization changes are not sufficiently convincing. Without a quantification of both contributions, which is absent in the manuscript, it is not clear which one dominates, and hence the claim that flexoelectricity dominates over piezoelectricity is not justified. The references pointing at similar effects mentioned in the authors' response and in the updated manuscript correspond to studies on materials that are structurally very different and may show a very different flexoelectric response than CuInP_2S_6 (BiFeO_3 , SrTiO_3 , sodium bismuth titanate–0.25 strontium titanate), so I believe they are not sufficient to justify this point. What's more, the piezoelectric coefficient of CuInP_2S_6 has been reported to be particularly large (Nano Energy 99, 107371), and the (homogeneous) strains induced by the PFM tip on a freestanding membrane should be rather large. Therefore, the piezoelectric contribution is expected to be large too.

Response:

We agree with the reviewer that we should provide more discussion about the coexistence of the piezoelectricity and flexoelectricity. We also apologize for the insufficient explanation on the [A] coexistence of the piezoelectricity and flexoelectricity and [B] its corresponding flexoelectric dominance.

[A] Coexistence of the \$\text{CuInP}_2\text{S}_6\$ piezoelectricity and \$\text{CuInP}_2\text{S}_6\$ flexoelectricity

According to the raised concerns, we have considered the coexistence of the CuInP_2S_6 piezoelectricity and CuInP_2S_6 flexoelectricity. During the tip-induced strain engineering of

2D van der Waals CuInP_2S_6 films, the piezoelectric field and flexoelectric field has been concurrently considered in previous CuInP_2S_6 research [Jiang, Xingan, et al. *Advanced Functional Materials* **33.40** (2023)]. We thank the reviewer for pointing out the insufficient explanations and have revised the sentence as follows.

[Revised Text] Main text (Page 7):

During the tip-induced strain engineering, the homogeneous strain (piezoelectricity) and the inhomogeneous strain (flexoelectricity) has been concurrently generated at the free-standing region. Hence, the coexistence of piezoelectric effect and flexoelectric effect should be considered.

[B] Observation of the flexoelectric dominance of CuInP_2S_6

Due to the insufficient explanation of flexoelectric dominance, we have inserted the additional supplementary figure, reference, and corresponding revisions into our revised manuscript to further clarify the flexoelectric dominance of nanogap CuInP_2S_6 . Jiang, Xingan, et al. calculated the cross-sectional distribution of CuInP_2S_6 flexoelectric fields and CuInP_2S_6 piezoelectric fields within applied tip force of 144 nN. Calculated field distribution completely can be divided into the piezoelectricity (homogeneous strain) and flexoelectricity (inhomogeneous strain). As the piezoelectricity refers the electrical polarization within homogeneous strain, the cross-sectional piezoelectric field distribution (**Fig. 4g** in Jiang, Xingan, et al. *Advanced Functional Materials* **33.40** (2023)) accurately correlates with the definition of piezoelectricity. While the flexoelectricity was locally generated from the lattice bending and its corresponding net polarization shift in specific lattice cell, the cross-sectional flexoelectric field simulation (**Fig. 5f**) exactly correlates with the definition of flexoelectricity and cross-sectional flexoelectric field distribution in suspended substrate (**Fig. 4f** in Jiang, Xingan, et al. *Advanced Functional Materials* **33.40** (2023)). To clarify the flexoelectric dominance in CuInP_2S_6 with nanoscale mechanical bending, flexoelectric field distribution in nanowrinkle has been observed in Supplementary

Fig. 6a, Fig. 6b, and Fig. 6c. As the nanowrinkle structure was generated without tip-induced vertical shear strain, internal polarization of nanowrinkle has been dominantly generated from the local lattice bending and its corresponding flexoelectricity. As shown in Supplementary Fig. 6c, nanowrinkle PFM amplitude line profile inversely correlates with calculated surface potential line profile (**Fig. 4h** in Jiang, Xingan, et al. *Advanced Functional Materials* **33.40** (2023)), while the nanowrinkle PFM phase accurately corresponds with calculated surface potential line profile (**Fig. 4h** in Jiang, Xingan, et al. *Advanced Functional Materials* **33.40** (2023)). As a result, the surface potential of flexoelectric field (**Fig. 4h** in Jiang, Xingan, et al. *Advanced Functional Materials* **33.40** (2023)) indicates the accurate corresponds with nanowrinkle CuInP_2S_6 PFM image (Supplementary Fig. 6b and Supplementary Fig. 6c). Hence, the flexoelectric dominance was clearly observed in nanoscale geometry with consideration of piezoelectric field. Moreover, tip-induced strain engineering (480 nN, 600 nN, 720 nN) has been conducted within substrate suspension to further minimize the flexoelectric effects. Within substrate suspension, applied tip force exhibits linear correlation with the distance (Supplementary Fig. 6d). Hence, the applied tip force (480 nN, 600 nN, 720 nN) is insufficient to activate the lattice bending in suspended substrate. In this case, lattice bending can be completely hindered, which effectively minimizes the flexoelectric field effect. To further confirm the substrate suspension effects, hysteresis variation was spatially mapped in suspended substrate with applied tip force (480 nN, 600 nN, 720 nN), which results in absence of hysteresis variation (Supplementary Fig. 6e). In conclusion, the flexoelectric dominance has been clarified with additional PFM experiments in suspended substrate and nanowrinkle structure. We thank the reviewer for pointing this out and have revised the original text with the additional explanation as follows:

[Revised Text] Main text (Page 7):

Nonetheless, the flexoelectricity is an inherently size-dependent phenomenon. When the flake thickness decreases to the nanoscale dimension, the inhomogeneous strain

dominantly overwhelms the homogeneous strain, which are simultaneously induced by tip-induced vertical shear strain (Supplementary Fig. 5). Additionally, cross-sectional field calculation [Jiang, Xingan, et al. *Advanced Functional Materials* **33.40** (2023)] accurately correlates with PFM image of CuInP_2S_6 nanowrinkle, CuInP_2S_6 nanogap (Fig. 2c), and free-standing flexoelectric simulation (Fig. 5f). To further clarify the flexoelectric dominance in CuInP_2S_6 with nanoscale mechanical bending, flexoelectric field distribution in nanowrinkle has been observed in Supplementary Fig. 6a, b, and c. As the nanowrinkle structure was generated without tip-induced vertical shear strain, internal polarization of nanowrinkle has been dominantly generated from the local lattice bending and its corresponding flexoelectricity. As shown in Supplementary Fig. 6b and c, nanowrinkle PFM amplitude inversely correlates with calculated surface potential of downward flexoelectricity [Jiang, Xingan, et al. *Advanced Functional Materials* **33.40** (2023)], while the nanowrinkle PFM phase linearly corresponds with surface potential of downward flexoelectricity [Jiang, Xingan, et al. *Advanced Functional Materials* **33.40** (2023)]. Moreover, tip-induced strain engineering has been conducted within substrate suspension to further minimize the flexoelectric effects. Within substrate suspension, applied tip force exhibits linear correlation with the distance (Supplementary Fig. 6d). Therefore, the applied tip force (480 nN, 600 nN, 720 nN) is insufficient to activate the lattice bending in suspended substrate. In this case, plastic deformation can be completely hindered, which effectively minimizes the flexoelectric field effect. The hysteresis variation was also spatially mapped in suspended substrate with each applied tip force, resulting the absence of hysteresis variation of suspended CuInP_2S_6 (Supplementary Fig. 6e). Hence, the flexoelectric dominance was clearly observed in nanoscale geometry, constructed as the nanowrinkle structure and the suspended substrate [Park, Sung Min, et al. *Nature nanotechnology* **13.5** (2018)] [Das, Saikat, et al. *Nature communications* **8.1** (2017)], [Molina-Luna, Leopoldo, et al. *Nature communications* **9.1** (2018)].

[Revised Text] Supplementary Information (Page 7):

Supplementary Fig. 6 has been added as follows.

Supplementary Fig. 6. Observation of flexoelectric dominance with nanoscale mechanical bending. Schematic of (a) upward lattice bending in nanowrinkle structure. Spatial distribution of (b) PFM responses and (c) its corresponding line profile, which exhibits the flexoelectric dominance in wrinkle apex area. (d) FD curve measurements at the suspended substrate. (e) PFM hysteresis mapping of substrate suspension, which effectively hinders the lattice bending and its corresponding hysteresis variation with each applied force.

[Revised Text] Supplementary Information (Page 12):

Supplementary Fig. 11 has been edited as follows.

Supplementary Fig. 11. Spatial PFM hysteresis mapping of nanoscale axial bending structure. (a) 3D topography of axial nanogap structure and PFM hysteresis mapping within (b) -5 V–5 V sweep range (ferro-ionic phase) and (c) -10 V–10 V sweep range (paraelectric phase). (d) 3D topography of axial nanowrinkle structure and PFM hysteresis mapping within (e) -10 V–10 V sweep range.

Reviewer (#3)'s COMMENTS:

the authors responded point by point in the revised version. as far as my comment part, the authors offered the effect of ferro-ionic conduction on the temperature under 315 K, 330K, 350 K, and 400 K conditions. from the results, it can be seen clearly that

temperature impacts the I-V curves in the temperature span from 315 to 400K, therefore, i think, ferro-ionic conductive behavior is temperature-dependent.

Response:

We thank the reviewer for the constructive comments to improve the completeness of our work. Based on the raised concerns, we have performed additional experiments and point-by-point responses with corresponding revisions to our revised manuscript.

Review #3_Q1)

however, the authors use the word "temperature-independence", i was confused, was it a mistake or not?

Response:

We thank the reviewer for pointing out our typos. To clearly deliver this information, we have thoroughly replaced the terminology "temperature-independent" to "temperature-dependent" in the revised manuscript.

[Revised Text] Main text (Page 11):

Because a temperature-dependent CuInP_2S_6 phase transition has been reported at 10^2 MV/m and 300 K [Zhou J, *et al. Advanced Materials*, 2302419 (2023)], the free-standing 2D nanogap device also exhibited temperature-dependent ferro-ionic conductive behavior (Supplementary Fig. 13) under the 400 K condition. Moreover, the IV hysteresis behavior in -10 V–10 V sweep range is inhomogeneous, which is attributed to the ferro-ionic phase transition. While the -10 V–10 V range possess the possibility of ferro-ionic phase transition within temperature variation, the -20 V–20 V sweep range clearly exceed the ferro-ionic conduction V_{th} .

[Revised Text] Supplementary Information (Page 14):

Supplementary Fig. 13 has been edited as follows.

Supplementary Fig. 13. Temperature effects in free-standing 2D ferroelectric nanogap device. (a) OM image and (b) 3D topography of free-standing 2D nanogap device, which is fabricated for temperature effects. Within various temperature conditions, ferro-ionic conduction mapping at (c) -10 V–10 V sweep range, and (d) -20 V–20 V sweep range, exhibiting the temperature-dependent conduction characteristics.

=====
We very appreciate the reviewers' all the valuable comments, which greatly helped us to improve this manuscript.
=====

=====
Additional Revision for Final Figure Set
=====

Furthermore, we have performed additional trivial revision to increase the visibility of final figure set, which effectively improves the completeness of this manuscript.

Additional Revision #1)

To increase the visibility, the pre-characterization of single flake (Supplementary Fig. 4 and Supplementary Fig. 5) has been merged in Supplementary Fig. 3 as follows.

[Revised Text] Supplementary Information (Page 4):

Supplementary Fig. 3 has been merged as follows.

Supplementary Fig. 3. Pre-characterization of cross-sectional atomic configuration and piezoelectric response in single CuInP₂S₆/ α -In₂Se₃ flake. (a) Cross-sectional EDS mapping, (b) SAED pattern, and (c) cross-sectional TEM image of single α -In₂Se₃ flake. (d) Cross-sectional EDS mapping, (e) SAED pattern, and (f) cross-sectional TEM image of single CuInP₂S₆ flake. (g) PFM amplitude and (i) PFM phase of α -In₂Se₃ single flake. (h) PFM amplitude and (j) PFM phase of CuInP₂S₆ single flake. High-resolution XPS spectra

of CuInP_2S_6 single flake, which consisted of (k) Cu 2p peak and (m) In 3d peak. High-resolution XPS spectra of $\alpha\text{-In}_2\text{Se}_3$ single flake, which deconvoluted with (i) In 3d peak, and (n) Se 3d peak.

Additional Revision #2)

To increase the visibility, the SEM image of nanogap structure (Supplementary Fig. 1b) has been replaced as follows.

[Revised Text] Supplementary Information (Page 2):

Supplementary Fig. 1 has been edited as follows.

Supplementary Fig. 1. Structural characterization of axial nanogap structure. (a)

Optical image of the axial nanogap structure, consisting of 500 nm width and 150 nm depth. (b) SEM image of fabricated axial nanogap array. (c) Optical image of free-standing 2D $\alpha\text{-In}_2\text{Se}_3/\text{CuInP}_2\text{S}_6$ heterojunction on the axial nanogap structure and (d) its

corresponding cross-sectional TEM image of free-standing 2D α - $\text{In}_2\text{Se}_3/\text{CuInP}_2\text{S}_6$ heterojunction. (e) Line profile and (f) its corresponding CAFM IV curve at the free-standing 2D ferroelectric heterojunction with various applied force, enabling the comparison with free-standing states (0 nN).

Additional Revision #3)

To increase the visibility, the PFM image of nanowrinkle (Fig. 2c) has been replaced as follows.

[Revised Text] Figure 2 (Page 28):

Fig. 2 has been edited as follows.

Fig. 2: Local flexoelectric domain mapping with local strain distribution. Schematic of the local strain distribution and its corresponding flexoelectric field in the (a) nanowrinkle

and (b) nanogap. Spatial domain imaging of the (c) nanowrinkle CuInP_2S_6 , (d) nanogap CuInP_2S_6 , (e) nanowrinkle In_2Se_3 , and (f) nanogap In_2Se_3 . Owing to the local flexoelectric polarization, PFM measurements indicate the downward polarization, as observed in the PFM amplitude and PFM phase. The electron distribution was correlatively obtained for the ferroelectric insulator CuInP_2S_6 and ferroelectric semiconductor In_2Se_3 , which originated from the heterogeneous carrier density of the semiconductor and insulator.

[Revised Text] Main text (Page 7):

Heterogeneous piezoresponse force microscopy (PFM) amplitude and phase at the edge and center areas were observed to be 11.37 mV / 159.89° (CuInP_2S_6 nanogap), 1.04 mV / 111.84° ($\alpha\text{-In}_2\text{Se}_3$ nanogap), 3.36 mV / 249.67° (CuInP_2S_6 nanowrinkle), and 11.37 mV / 356.30° ($\alpha\text{-In}_2\text{Se}_3$ nanowrinkle).

Additional Revision #4)

To clarify the device structure, the “memristor cross-bar” has been replaced to “cross-bar array” (Fig. 1c) as follows.

Fig. 1: Universal axial nanogap platform for programmable flexoelectric energy conversion. (a) Schematic of free-standing single 2D ferro-ionic memristor for spatial confinement of conductive filaments. (b) Free-standing 2D α -In₂Se₃/CuInP₂S₆ heterostructure and its corresponding TEM cross-sectional image and SAED mapping. (c) Schematic illustration of the 2D ferro-ionic memristor structure, constructed with an Au bottom electrode/ α -In₂Se₃/CuInP₂S₆/Pt top electrode. (d) SEM image of the axial nanogap array with a 500 nm width and 150 nm depth. (e) Cross-sectional 3D line profile with tip-induced strain engineering. Flexoelectric current amplification effects in the (f) full I-V curve through shear strain.

REVIEWERS' COMMENTS

Reviewer #2 (Remarks to the Author):

The authors have responded to my remaining questions.

Reviewer #3 (Remarks to the Author):

the mistake was corrected.

[Revision #3] Response to the *Nature Communications* Reviewers' Comments

Dear reviewers,

We the authors deeply thank the reviewers for their thorough reading of our manuscript and the constructive comments. With regard to the reviewers' concerns raised, we have revised the manuscript and supporting information based on the reviewers' comments.

The reviewers' comments appear in black, and the authors' responses in blue. In the revised manuscript, the changes with respect to the previous version are highlighted.

Sincerely,

Taesung Kim

=====

Point-by-point Response

=====

Reviewer (#2)'s COMMENTS:

The authors have responded to my remaining questions.

Response:

We really appreciate the "reviewer #2" for the constructive comments to improve our manuscript. We are glad that the reviewer finds improved qualities of our manuscript and recommends the publications.

Reviewer (#3)'s COMMENTS:

the mistake was corrected.

Response:

We thank the "reviewer #3" for the helpful comments to improve our manuscript. We are glad that the reviewer satisfactorily reads our answer and finds our manuscript suitable for publication.